# Scalable Second-order Riemannian Optimization for $K$-means Clustering

**Peng Xu**[*]
University of Illinois Urbana-Champaign
pengxu1@illinois.edu

**Chun-Ying Hou**[*]
University of Illinois Urbana-Champaign
cyhou2@illinois.edu

**Xiaohui Chen**
University of Southern California
xiaohuic@usc.edu

**Richard Y. Zhang**
University of Illinois Urbana-Champaign
ryz@illinois.edu

## Abstract

Clustering is a hard discrete optimization problem. Nonconvex approaches such as low-rank semidefinite programming (SDP) have recently demonstrated promising statistical and local algorithmic guarantees for cluster recovery. Due to the combinatorial structure of the $K$-means clustering problem, current relaxation algorithms struggle to balance their constraint feasibility and objective optimality, presenting tremendous challenges in computing the second-order critical points with rigorous guarantees. In this paper, we provide a new formulation of the $K$-means problem as a smooth unconstrained optimization over a submanifold and characterize its Riemannian structures to allow it to be solved using a second-order cubic-regularized Riemannian Newton algorithm. By factorizing the $K$-means manifold into a product manifold, we show how each Newton subproblem can be solved in linear time. Our numerical experiments show that the proposed method converges significantly faster than the state-of-the-art first-order nonnegative low-rank factorization method, while achieving similarly optimal statistical accuracy.

## 1 Introduction

Clustering is a cornerstone of modern unsupervised learning, where the goal is to group similar observations into meaningful clusters. The problem is commonly approached through the $K$-means formulation, which seeks to partition $n$ data points $X_1, X_2, \ldots, X_n \in \mathbb{R}^d$ into $K$ disjoint groups $G_1, \ldots, G_K$ by maximizing the total intra-cluster similarity:

$$\max_{G_1,\ldots,G_K} \left\{ \sum_{k=1}^{K} \frac{1}{|G_k|} \sum_{i,j \in G_k} \langle X_i, X_j \rangle : \bigsqcup_{k=1}^{K} G_k = [n] \right\}. \tag{1}$$

Here, the inner product $\langle X_i, X_j \rangle = X_i^\top X_j$ is used to measure pairwise similarity, $|G_k|$ denotes the cardinality of $G_k$, and $\sqcup$ denotes disjoint union. Most common algorithms for $K$-means clustering, including Lloyd's algorithm (Lloyd, 1982) and spectral clustering (Ng et al., 2001; von Luxburg, 2007), can be understood as heuristics for finding "good enough" solutions to the discrete optimization (1). These methods do not come with any guarantees of local optimality, let alone global optimality. Indeed, it is commonly argued that globally solving (1) is NP-hard in the worst-case (Dasgupta, 2007; Aloise et al., 2009), and would lead to statistically meaningless clustering that overfits the data.

Yet in average-case regimes, globally solving the $K$-means optimization problem (1) can be both computationally tractable as well as statistically optimal. In particular, when the data $X_1, \ldots, X_n$ arise from a Gaussian mixture model with sufficiently well-separated components, Chen & Yang (2021b) showed that a well-known semidefinite programming (SDP) relaxation of Peng & Wei (2007),

---

[*]These authors contributed equally.

written

$$\max_{Z \in \mathbb{R}^{n \times n}} \left\{ \langle XX^\top, Z \rangle + \mu \sum_{i,j} \log(Z_{i,j})_+ : Z\mathbf{1}_n = \mathbf{1}_n, \, \text{tr}(Z) = K, \, Z \succeq 0 \right\}, \qquad (2)$$

where $X = [X_1, \ldots, X_n]^\top$ and $\log(Z_{i,j})_+ := \log(\max\{Z_{i,j}, 0\})$, is guaranteed to compute the globally optimal clusters $G_1^\star, \ldots, G_K^\star$ for (1) in the limit $\mu \to 0^+$ in polynomial time, that in turn recover the ground truth partitions. Note that the formulation (2) is equivalent to the standard $K$-means SDP formulation with the elementwise nonnegativity constraint $Z_{i,j} \geq 0$ in Peng & Wei (2007); Chen & Yang (2021b) (see Appendix A for more discussions). Moreover, this recovery occurs as soon as the separation between the clusters is large enough for it to be possible. Put in another way, if solving (2) does not recover the ground truth partitions, then the clusters are too closely spaced in a way that makes recovery inherently impossible in an information-theoretic limit sense, see Section 2.1 for more details.

Unfortunately, the SDP (2) is not a practical means of solving (1) to global optimality, due to its need to optimize over an $n \times n$ matrix to cluster $n$ samples. Following Burer & Monteiro (2003); Boumal et al. (2020), a natural alternative is to factor $Z = UU^\top$ into its $n \times r$ factor matrix $U$ for rank parameter $r \geq K$, impose the logarithmic penalty over $U$ instead of $Z$, and then directly optimize over $U$:

$$\max_{U \in \mathbb{R}^{n \times r}} \left\{ \langle XX^\top, UU^\top \rangle + \mu \sum_{i,j} \log(U_{i,j})_+ : UU^\top \mathbf{1}_n = \mathbf{1}_n, \, \text{tr}(UU^\top) = K \right\}. \qquad (3)$$

This reduces the number of variables and constraints from $O(n^2)$ down to $O(n)$, but at the cost of giving up the convexity of the SDP. In general, we can at best hope to compute critical points, which may be spurious local minima or saddle points. The core motivation for our approach, and the impetus for this paper, is the surprising empirical observation that all second-order critical points are global optima in this setting; this is formalized as the following assumption.

**Assumption 1** (Benign nonconvexity). *In the average-case regime when (2) globally solves (1), all approximate second-order critical points in (3) are within a neighborhood of a global optimum.*

The phenomenon of benign nonconvexity is well-documented in the *unconstrained* version—optimizing over semidefinite $Z \succeq 0$ by factorizing $Z = UU^\top$—dating back to the early works of Burer & Monteiro (2003). In contrast, it is rarely seen in our *nonnegative* variant, which adds the elementwise constraint $U \geq 0$ to enforce doubly nonnegativity in $Z = UU^\top$. Despite a superficial similarity, the two formulations differ in fundamental ways, with the nonnegative case known to admit numerous spurious critical points; see Section 1.2 for some classic and recent examples. Nevertheless, we consistently observe that all second-order critical points correspond to global optima, that in turn successfully recover the optimal clusters.

### 1.1 CONTRIBUTIONS: CHEAP AND FAST CONVERGENCE TO SECOND-ORDER CRITICAL POINTS

Under Assumption 1, globally solving the $K$-means optimization problem (1) reduces to that of computing a second-order critical point for (3). Unfortunately, in the constrained nonconvex setting, there is no general-purpose algorithm that is rigorously guaranteed to compute a second-order critical point. The core issue is the need to maintain *feasibility*, i.e. for each iterate $U$ to satisfy the nonconvex constraints $UU^\top \mathbf{1}_n = \mathbf{1}_n$ and $\text{tr}(UU^\top) = K$, while making progress towards optimality. General-purpose solvers like `fmincon` (Byrd et al., 2000) and `knitro` (Byrd et al., 2006) promise convergence only to critical points of an underlying merit function, which may be infeasible for the original problem. Augmented Lagrangian methods guarantee convergence only to first-order critical points, and only when starting within a local neighborhood (Zhuang et al., 2024). This is a significant departure from the unconstrained nonconvex setting, where a diverse range of algorithms—both cheap first-order algorithms like gradient descent, as well as rapidly-converging second-order methods like trust-region Newton's method—globally converge to a second-order critical point starting from any initial point.

Our first contribution is to present an interpretation of (3) as a *smooth unconstrained optimization over a Riemannian manifold*. This allows the immediate benefit of extending the wide array of

unconstrained optimization algorithms to the constrained setting, as well as their accompanying guarantees for first- and second-order optimality. For the first time in the context of $K$-means, we open the possibility to guarantee global convergence to first- and second-order optimality.

Our second contribution is to show that *second-order Riemannian algorithms can be implemented with linear per-iteration costs* with respect to the number of samples $n$. In other words, of all practical algorithms to compute second-order critical points, we show that the one with the best iteration complexity (second-order methods) can be improved to have the same per-iteration costs as first-order methods. Our final algorithm computes $\epsilon$ second-order points in $n \cdot \epsilon^{-3/2} \cdot \text{poly}(r, d)$ time.

## 1.2 RELATED WORK

Benign nonconvexity in the unconstrained Burer–Monteiro factorization $Z = UU^\top$ has been empirically observed since the early 2000s (Burer & Monteiro, 2003), and widely exploited in nonconvex low-rank algorithms in machine learning. In the past decade, theory has been developed to explain this phenomenon under some specialized settings (Bhojanapalli et al., 2016; Ge et al., 2016; Bandeira et al., 2016; Boumal et al., 2016; Ge et al., 2017). Unfortunately, these guarantees tend to be conservative in the number of samples or the level of noise; they capture the general phenomenon but cannot rigorously explain what is broadly observed in practice.

In contrast, the *nonnegative* Burer–Monteiro factorization $Z = UU^\top$ with $U \geq 0$ is widely understood *not* to exhibit benign nonconvexity. To give two simple examples, the functions $f(U) = \langle SU, U \rangle$ and $f(U) = \|UU^\top - U_\star U_\star^\top\|_F^2$ are easily confirmed to exhibit benign nonconvexity over $U \in \mathbb{R}^{n \times r}$. But imposing $U \geq 0$ causes spurious local minima to proliferate; this is unsurprising because both problems, namely copositive testing (Murty & Kabadi, 1987) and complete positive testing (Dickinson & Gijben, 2014), are well-known to be NP-hard. For a more sophisticated example, the function $f(U) = \|\mathcal{A}(UU^\top - U_\star U_\star^\top)\|^2$ is well known to exhibit benign nonconvexity when the linear operator $\mathcal{A} : S^n(\mathbb{R}) \to \mathbb{R}^m$ satisfies the restricted isometry property (RIP) (Bhojanapalli et al., 2016). In this context, a recent work (Zhang, 2025) gave a strong counterexample for the equivalent statement over $U \geq 0$.

Therefore, even though $K$-means is widely known to admit a nonnegative Burer–Monteiro reformulation (Peng & Wei, 2007), there have been only two prior works that actually follow this approach, to the best of our knowledge. Neither of these can rigorously guarantee global optimality under Assumption 1. The first is the first-order Riemannian method introduced by Carson et al. (2017). It solves the following:

$$\min_{U \in \mathcal{M}'} \left\{ -\langle XX^\top, UU^\top \rangle + \lambda \|U_-\|_F^2 \right\} \tag{4}$$

where $\mathcal{M}' := \left\{ U \in \mathbb{R}^{n \times K} : U^\top U = I_K, UU^\top \mathbf{1}_n = \mathbf{1}_n \right\}$, and $U_- = \max\{-U, 0\}$ is the (entrywise) negative part of $U$, $\lambda \geq 0$ is the penalty parameter for $U \geq 0$. Although superficially similar, their approach fundamentally lacks a convergence guarantee to a second-order critical point, due to: (i) their nonsmooth objective; (ii) their use of a smooth penalty, which cannot truly enforce feasibility $U \geq 0$; (iii) their use of a first-order method, which can get trapped at a saddle point. Moreover, their manifold is geometrically complicated, necessitating an expensive retraction that costs $O(n^2)$ time, which prevents their method from scaling to large datasets.

The second work is the nonnegative low-rank (NLR) method of Zhuang et al. (2024). This is a simple projected gradient descent that directly projects $U$ onto the nonnegative spherical constraint and deals the row sum constraint $UU^\top \mathbf{1}_n = \mathbf{1}_n$ via the augmented Lagrangian method. It is a first-order primal-dual method that can only achieve local linear convergence in a neighborhood of its global solution. Like Carson et al. (2017), it is unclear whether there is a pathway that this algorithm can lead to a global optimality guarantee, or even to second-order optimality.

Recently, hybrid methods (Wang & Hu, 2025; Monteiro et al., 2025; Hou et al., 2025; Wang et al., 2023) have been proposed for low-rank SDPs. These approaches handle simple manifold constraints via Riemannian optimization, while enforcing the remaining constraints through augmented Lagrangian updates. In contrast, our reformulated $K$-means problem requires strict feasibility with respect to both nonnegativity and the simplex-type manifold constraints, since the recovered factor must encode a valid partition. Augmented Lagrangian or projection-based schemes do not preserve this property and would break the structural guarantees on which our method relies.

## 2 BACKGROUND

### 2.1 SDP RELAXATION OF $K$-MEANS

Despite the worst-case NP-hardness of the $K$-means clustering optimization problem (1), common practical heuristics and relaxed formulations like Lloyd's algorithm (Lloyd, 1982), spectral clustering (Ng et al., 2001; von Luxburg, 2007), nonnegative matrix factorization (NMF) (He et al., 2011; Kuang et al., 2015; Wang & Zhang, 2012) and SDPs (Peng & Wei, 2007; Royer, 2017; Fei & Chen, 2018; Chen & Yang, 2021a) work surprisingly well at solving it for real-world data. To explain this discrepancy between theory and practice, suppose that the data $X_1, \ldots, X_n \in \mathbb{R}^d$ are generated from a standard Gaussian mixture model (GMM)

$$X_i = \mu_k + \varepsilon_i, \qquad \varepsilon_i \overset{\text{i.i.d.}}{\sim} \mathcal{N}(0, \sigma^2 I_d), \quad \text{for } i \in G_k^*, \tag{5}$$

where $G_k^*$ denotes the ground truth clusters. Chen & Yang (2021b) proved that the SDP (2) of Peng & Wei (2007) (as $\mu \to 0^+$) achieves a *sharp phase transition* on the separation of centroids for the clustering problem, in any dimension $d$ and sample size $n$. Let

$$\overline{\Theta}^2 := 4\sigma^2 \left( 1 + \sqrt{1 + \frac{Kd}{n \log n}} \right) \log n, \tag{6}$$

and $\Theta_{\min} := \min_{1 \le j < k \le K} \|\mu_j - \mu_k\|$ be the minimum centroid separation. Assume that $m = n/K$ is an integer without loss of generality and consider any $\alpha > 0$. As soon as the exact recovery becomes possible in the regime $\Theta_{\min} \ge (1+\alpha)\overline{\Theta}$, the SDP approach (2) solves the $K$-means problem without clustering error with high probability. For precise statements on the information-theoretic threshold, please refer to Theorem 3 in Appendix C. As an immediate consequence of the global optimality guarantee of the $K$-means SDP in (2), we deduce that the global solution of the nonconvex low-rank SDP in (3) solves the $K$-means clustering problem in (1) in the exact recovery regime.

Next, from the membership matrix $Z$, we would like to convert it to the cluster label.

**Lemma 1.** *Let $Z = Z^\top \in \mathbb{R}^{n \times n}$ be the symmetric block-diagonal matrix defined by $Z_{ij} = 1/|G_k|$ if $i, j \in G_k$, and $Z_{ij} = 0$ otherwise. Then for any integer $r \in [K, n]$, there is a unique (up to column permutation) $U \in \mathbb{R}_+^{n \times K}$ such that $Z = UU^\top$. Moreover, $U$ can be recovered from any $\hat{U} \in \mathbb{R}^{n \times r}$ satisfying $Z = \hat{U}\hat{U}^\top$ in $n \cdot \mathrm{poly}(r)$ time.*

For each block-diagonal membership matrix $Z$, the unique $U \in \mathbb{R}_+^{n \times K}$ in Lemma 1 is the associated group assignment matrix, i.e. the $k$-th column of $U$ provides a one-hot encoding of membership in the $k$-th cluster.

### 2.2 CRITICAL POINTS IN CONSTRAINED OPTIMIZATION

The problems considered in this paper are instances of the following

$$\min_{U \in \mathcal{M}} f(U), \qquad \mathcal{M} = \{U \in \mathbb{R}^{n \times r} : \mathcal{A}(UU^\top) + \mathcal{B}(U) = c\}, \tag{7}$$

where the linear operators $\mathcal{A} \colon \mathbb{R}^{n \times n} \to \mathbb{R}^m$ and $\mathcal{B} \colon \mathbb{R}^{n \times r} \to \mathbb{R}^m$ and right-hand side $c \in \mathbb{R}^m$ together are assumed to satisfy the linear independence constraint qualification (LICQ)

$$2[\mathcal{A}^\top(y)]U + \mathcal{B}^\top(y) = \mathbf{0} \iff y = \mathbf{0} \quad \forall U \in \mathcal{M}. \tag{8}$$

In this context, $U \in \mathbb{R}^{n \times r}$ is said to be *feasible* if it satisfies $U \in \mathcal{M}$. The feasible point $U$ is an $\epsilon$-first-order critical point if it satisfies

$$\text{exists} \quad y \in \mathbb{R}^m \quad \text{s.t.} \quad \left\| \nabla f(U) + 2[\mathcal{A}^\top(y)]U + \mathcal{B}^\top(y) \right\| \le \epsilon, \tag{9}$$

and an $\epsilon$-second-order critical point if it additionally satisfies

$$\langle \nabla^2 f(U) + 2[\mathcal{A}^\top(y)], \dot{U}\dot{U}^\top \rangle \ge \sqrt{\epsilon} \|\dot{U}\|^2 \quad \forall \dot{U} \in \mathrm{T}_U \mathcal{M} \tag{10}$$

over the *tangent space* of $\mathcal{M}$ at the point $U$, given by $\mathrm{T}_U \mathcal{M} = \{\dot{U} \in \mathbb{R}^{n \times r} : \mathcal{A}(\dot{U}U^\top + U\dot{U}^\top) + \mathcal{B}(\dot{U}) = 0\}$. Under LICQ (8), every local minimum (and hence the global minimum) is guaranteed to be an $\epsilon$-second-order critical point (for any $\epsilon \ge 0$). Unfortunately, there is no general-purpose algorithm that is guaranteed to converge to a critical point, due to the need to achieve and maintain feasibility across all iterates.

## 2.3 SECOND-ORDER RIEMANNIAN OPTIMIZATION

Riemannian algorithms are special algorithms that maintain feasible iterates through a problem-specific *retraction* operator, and are hence able to rigorously guarantee convergence to critical points. The basic idea is to improve a feasible iterate $U \in \mathcal{M}$ by tracing a smooth curve on the feasible set $\gamma : [0, \epsilon) \to \mathcal{M}$ that begins at $\gamma(0) = U$ and proceeds in a direction of descent $\dot{\gamma}(0) = \dot{U} \in \mathrm{T}_U \mathcal{M}$. In analogy with unconstrained algorithms, a good choice of $\dot{U} \in \mathrm{T}_U \mathcal{M}$ is found through a local Taylor expansion

$$f\big(\gamma(t)\big) = f(U) + t\langle \mathrm{grad}\, f(U), \dot{U}\rangle + \frac{t^2}{2}\langle \mathrm{Hess}\, f(U)[\dot{U}], \dot{U}\rangle + O(t^3), \tag{11}$$

where $\mathrm{grad}\, f$ and $\mathrm{Hess}\, f$ are respectively the *Riemannian gradient* and *Riemannian Hessian* of $f$ on the manifold $\mathcal{M}$. Afterwards, we trace the curve $\gamma(t) = \mathrm{R}_U(t\dot{U})$ using a *second-order retraction* operator $\mathrm{R}_U \colon \mathrm{T}_U \mathcal{M} \to \mathcal{M}$ satisfying

$$\mathrm{R}_U(0) = U, \qquad \frac{d}{dt}\mathrm{R}_U(t\dot{U})\Big|_{t=0} = \dot{U}, \qquad \frac{d^2}{dt^2}\mathrm{R}_U(t\dot{U})\Big|_{t=0} \perp \mathrm{T}_U \mathcal{M},$$

for all $U \in \mathcal{M}$ and all $\dot{U} \in \mathrm{T}_U \mathcal{M}$. After choosing step-size $t$ so that $U_{\mathrm{new}} = \gamma(t)$ makes a sufficient improvement over $U$, we repeat the algorithm until it reaches an $\epsilon$-second-order critical point satisfying $\|\mathrm{grad}\, f(U)\| \leq \epsilon$ and $\lambda_{\min}\big(\mathrm{Hess}\, f(U)\big) \geq -\sqrt{\epsilon}$, which incidentally corresponds exactly to (9) and (10). Proofs for the following convergence result can be found in Zhang & Zhang (2018); Boumal et al. (2019); Agarwal et al. (2021); we have chosen the simplest but most restrictive settings to ease the exposition.

**Theorem 1** (Riemannian cubic-regularized Newton). *Suppose that $\min_{U \in \mathcal{M}} f(U) > -\infty$, and that the pullback $\hat{f} = f \circ \mathrm{R}_U$ has Lipschitz continuous Hessian for all $U \in \mathcal{M}$. Then, there exists a sufficiently large regularizer $L$ such that $U_{k+1} = \mathrm{R}_{U_k}(\dot{U}_k)$ where*

$$\dot{U}_k = \underset{\dot{U} \in \mathrm{T}_U \mathcal{M}}{\arg\min}\, f(U) + \langle \mathrm{grad}\, f(U), \dot{U}\rangle + \frac{1}{2}\langle \mathrm{Hess}\, f(U)[\dot{U}], \dot{U}\rangle + \frac{L}{6}\|\dot{U}\|^3$$

*converges to an $\epsilon$-second order critical point in $O(\epsilon^{-3/2})$ iterations, independent of dimension.*

Each iteration of Riemannian cubic-regularized Newton solves an expensive Newton subproblem. Although it converges in far fewer iterations compared to gradient methods, it is practically competitive only when the added cost of solving the Newton subproblem can be offset by the corresponding reduction in iteration count.

## 3 FORMULATION AND SOLUTION OF $K$-MEANS AS MANIFOLD OPTIMIZATION

We now explain how we solve (3) using a Riemannian optimization approach. As a first attempt, we can indeed verify that the the constraint set in (3), written

$$\mathcal{M} \coloneqq \mathcal{M}_r = \big\{U \in \mathbb{R}^{n \times r} : UU^\top \mathbf{1}_n = \mathbf{1}_n,\ \mathrm{tr}(UU^\top) = K\big\}, \tag{12}$$

is a manifold by checking that (8) holds (cf. Lemma 3 in the appendix). In fact, directly applying Riemannian optimization techniques results in a $K$-means algorithm very similar to the one proposed in Carson et al. (2017). The immediate and critical difficulty with this approach is the lack of an efficient retraction operator, which must be called at every iteration to keep iterates feasible $U \in \mathcal{M}$. For example, Carson et al. (2017) used a complicated exponential retraction that costs $O(n^2)$ time, hence bottlenecking the entire algorithm and preventing it from scaling to large $n$.

Instead, our first contribution in this paper is to reformulate (3) by establishing a submersion from the product manifold $\widetilde{\mathcal{M}} = \mathcal{V} \times \mathcal{O}_r$ to $\mathcal{M}$, where

$$\mathcal{V} = \Big\{V \in \mathbb{R}^{n \times (r-1)} : \mathbf{1}_n^\top V = 0,\ \mathrm{tr}(VV^\top) = K - 1\Big\},$$

$$\mathcal{O}_r = \big\{Q \in \mathbb{R}^{r \times r} : QQ^\top = I_r\big\}.$$

In words, $\mathcal{V}$ is a projected hypersphere and $\mathcal{O}_r$ is the set of $r \times r$ orthonormal matrices.

**Theorem 2.** *Let $\varphi(V, Q) := \hat{V}Q$, where $\hat{V} := \begin{bmatrix} \hat{\mathbf{1}}_n & V \end{bmatrix}$ with $\hat{\mathbf{1}}_n := n^{-1/2}\mathbf{1}_n$. Then $\mathcal{M} = \varphi(\widetilde{\mathcal{M}})$. Moreover, the Jacobian $\mathrm{D}\,\varphi \colon \mathrm{T}\,\widetilde{\mathcal{M}} \to \mathrm{T}\,\mathcal{M}$ is surjective for all $(V, Q) \in \widetilde{\mathcal{M}}$, i.e., $\varphi$ is a submersion.*

Having established the submersion property of $\varphi$, it is a standard result that every $\epsilon$-second order point of $\widetilde{\mathcal{M}}$ is also a $c\epsilon$-second order point on $\mathcal{M}$ for some constant rescaling factor $c$; see e.g. Example 3.14 and the surrounding text in Levin et al. (2025). Therefore, to solve (3), we equivalently solve

$$\min_{(V, Q) \in \widetilde{\mathcal{M}}} \langle C, VV^\top \rangle - \mu \sum_{i,j} \log\big(\varphi_{i,j}(V, Q)\big)_+, \tag{13}$$

where $C = -XX^\top$ is the (negative) data Gram matrix, and $\varphi_{i,j}$ is the $(i, j)$-th element of the operator $\varphi$ in Theorem 2. A basic but critical benefit of the reformulation (13) is that the product manifold $\widetilde{\mathcal{M}}$ admits a simple second-order retraction via its Euclidean projection (Boumal, 2023, Sec. 5.12)

$$\mathrm{R}_{(V,Q)}(\dot{V}, \dot{Q}) = \begin{bmatrix} \mathrm{Proj}_{\mathcal{V}}(V + \dot{V}) & \mathrm{Proj}_{\mathcal{O}_r}(Q + \dot{Q}) \end{bmatrix},$$

where

$$\mathrm{Proj}_{\mathcal{V}}(V) = \sqrt{K - 1}\,\frac{V - n^{-1}\mathbf{1}_n\mathbf{1}_n^\top V}{\|V - n^{-1}\mathbf{1}_n\mathbf{1}_n^\top V\|} \quad \text{and} \quad \mathrm{Proj}_{\mathcal{O}_r}(Q) = (QQ^\top)^{-1/2}Q.$$

It is easy to check that the retraction above costs just $O(nr + r^3)$ time to evaluate. In Section E.2, we give explicit expressions for the Riemannian gradient and Hessian and explain how they can be computed in $O(nr + r^3)$ time.

The appearance of the logarithmic penalty in (13) presents two difficulties. First, as a practical concern, any algorithm for (13) must begin at a *strictly feasible* point $(V_0, Q_0) \in \widetilde{\mathcal{M}}$ that additionally satisfies $\varphi(V_0, Q_0) > 0$. In Section E.4, we provide a good strictly feasible initial point, and prove that points exist only if the search rank is over-parameterized as $r > K$. Second, some special care is needed to rigorously apply the guarantees from Section 2.3, given that the penalty $\varphi(V, Q)$ is Lipschitz only when restricted to a closed and strictly feasible subset; see Section E.5 for details.

Together, these ingredients allow us to apply Riemannian gradient descent (Boumal et al., 2019) to (13) to compute an $\epsilon$-first-order critical point in $(n/\epsilon) \cdot \mathrm{poly}(r, d, K)$ time. In practice, the algorithm often converges to an $\epsilon$-second-order critical point, though this is not rigorously guaranteed without a carefully-tuned noise perturbation. Alternatively, we can apply the conjugate-gradients (CG) variant of the Riemannian trust-region algorithm (RTR), a general-purpose solver available in packages like `MANOPT` (Boumal et al., 2014) or `PYMANOPT` (Townsend et al., 2016), to guarantee convergence to an $\epsilon$-second-order critical point. Unfortunately, in our experiments, we observed that all of these algorithms experience unsatisfactorily slow convergence, due to the severe ill-conditioning introduced by the logarithmic penalty.

Instead, our best numerical results were obtained by the Riemannian cubic-regularized Newton (Theorem 1). Our key insight is that the algorithm can be implemented with just $O(nr^3)$ time per-iteration, by exploiting the underlying block-diagonal-plus-low-rank structure of the Riemannian Hessian. To explain, our core difficulty is to efficiently solve the Newton subproblem

$$\min_{Ap=0} g^\top p + \frac{1}{2}p^\top Hp + \frac{L}{6}\|p\|^3, \tag{14}$$

where $g$ and $H$ denote the vectorized Riemannian gradient and Hessian respectively, and $A$ implements the tangent space constraint $(\dot{V}, \dot{Q}) \in \mathrm{T}_{(V,Q)}\mathcal{M}$. We can verify that the subproblem contains $n(r - 1) + r^2 = O(nr)$ variables and is subject to $m = r + r(r + 1)/2 = O(r^2)$ constraints. Given that the subproblem has only linear constraints, its local minima must always satisfy the first- and second-order optimality conditions (9) and (10), which read

$$\begin{bmatrix} H + \lambda I & A^\top \\ A & 0 \end{bmatrix}\begin{bmatrix} p \\ q \end{bmatrix} = \begin{bmatrix} -g \\ 0 \end{bmatrix}, \quad \lambda = \frac{L}{2}\|p\|, \quad \xi^\top(H + \lambda I)\xi \geq 0 \text{ for all } \xi \text{ satisfying } A\xi = 0.$$

The following result can be viewed as a Riemannian extension of known results on the approximate minimization of cubic-regularized subproblems. It shows that, with sufficient regularization $L$, the global minimum corresponds to the unique second-order critical point.

**Lemma 2.** *Let $A$ have full row-rank (i.e. $AA^\top \succ 0$) and let $\lambda_{\min} = \min_{\|\xi\|=1, A\xi=0} \xi^\top H \xi$. For $\lambda > -\lambda_{\min}$, the parameterized solution*

$$p(\lambda) = \begin{bmatrix} I \\ 0 \end{bmatrix}^\top \begin{bmatrix} H + \lambda I & A^\top \\ A & 0 \end{bmatrix}^{-1} \begin{bmatrix} -g \\ 0 \end{bmatrix}$$

*is well-defined and $\|p(\lambda)\|$ is monotonously decreasing with respect to $\lambda$.*

The same lemma also suggests solving the Newton subproblem by simple bisection search, cf. (Cartis et al., 2011, Sec. 6.1). Indeed, the solution is just $p(\lambda_{\text{opt}})$, where $\lambda_{\text{opt}}$ is the solution to the *monotone* equation $2\lambda = L\|p(\lambda)\|$ (via Lemma 2). Thus, we pick a very small $\lambda_{\text{lb}} \approx -\lambda_{\min}$ such that $\|p(\lambda_{\text{lb}})\| > 2\lambda_{\text{lb}}/L$, a very large $\lambda_{\text{ub}}$ such that $2\lambda_{\text{ub}}/L > \|p(\lambda_{\text{ub}})\|$, and then perform bisection until $2\lambda_{\text{opt}} = L\|p(\lambda_{\text{opt}})\|$ is approximately found. For each $\lambda$, if $2\lambda < L\|p(\lambda)\|$, then we increase $\lambda$; otherwise, we decrease $\lambda$.

The main cost of the bisection search is the computation of $p(\lambda)$, which naively costs $O(n^3 r^3)$ time. For our specific problem, we explain in Section F how a block-diagonal-plus-low-rank structure in the Hessian $H$ reduces the computation cost to just $n \cdot \text{poly}(r, d)$ time. Applying Theorem 1 shows that the overall method computes an $\epsilon$-second-order critical point in $(n/\epsilon^{1.5}) \cdot \text{poly}(r, d, K)$ time.

## 4 NUMERICAL RESULTS

In this section, we showcase the superior performance of our proposed Riemannian second-order method for clustering on both synthetic Gaussian mixture models (GMM) and real-world mass cytometry (CyTOF) datasets. Compared to existing state-of-the-art methods, such as the nonnegative low-rank (NLR) factorization (Zhuang et al., 2024) and prior Riemannian $K$-means algorithms (Carson et al., 2017), our approach achieves faster convergence, higher clustering accuracy, and more reliable recovery of ground-truth cluster memberships. These results highlight the convergence and accuracy advantages of second-order methods when they can be implemented with per-iteration costs of just $O(n)$ time. The implementation details are deferred to Section H.

**Datasets.** We conducted experiments on both synthetic and real datasets. The synthetic data was generated from a standard $K$-component, $d$-dimensional Gaussian mixture model (GMM), with centroids placed at simplex vertices such that their separation equals $\gamma\overline{\Theta}^2$, where $\overline{\Theta}$ is the information-theoretic threshold for exact recovery in (6), and $\gamma$ controls separation. The real dataset came from mass cytometry (CyTOF) (Levine et al., 2015; Weber, 2015). It consists of 265,627 cell protein expression profiles across 32 markers, labeled into 14 gated cell populations. Following Zhuang et al. (2024), we uniformly sample 1,800 cells from $K = 4$ unbalanced clusters (labels 2, 7, 8, 9) from individual 1 for our experiment.

**Global optimality at second-order critical points (validation of Assumption 1).** Figure 1 shows the convergence behaviors of loss function (13) for GMMs ($n = 500, \gamma = 1.2, \mu = 0.01$) with 50 randomized initializations. We consistently observe that: (i) the loss value steadily decreases over iterations and converges rapidly near the globally optimal point; (ii) the Riemannian gradient norm dynamics suggest that our algorithm initially attempts to escape saddle points (with increased gradient norm) and eventually converges to second-order local optimality, where zero-loss is achieved, indicating global optimality. To verify second-order local optimality, we also plot the minimum eigenvalue of the Riemannian Hessian. This provides strong numerical evidence that near-second-order critical points are near-globally optimal, as posited by Assumption 1.

**Benchmark on real world data.** Prior studies on mass cytometry (CyTOF) and computer vision (CIFAR-10) datasets identified the nonnegative low-rank (NLR) factorization (Zhuang et al., 2024) as the most reliable clustering solver, attaining the lowest average mis-clustering error and the tightest variance compared to classical baselines such spectral clustering (SC), nonnegative matrix factorization (NMF), and $K$-means++ (Arthur & Vassilvitskii, 2007) ($KM$++). Our algorithm optimizes the same nonnegative low-rank model, so it inherits this reliability. Because it applies second-order Hessian updates rather than first-order gradients, it refines each iterate more thoroughly and therefore recovers the ground-truth membership matrix more accurately. Figure 3 illustrates this on CyTOF: both methods keep mis-clustering near zero, yet our solver achieves a smaller Frobenius gap to the oracle solution. The experiment was repeated 50 times on random subsamples of size $n = 1800$. An additional experiment on the CIFAR10 dataset can be found in Section G.

**Comparison with NLR.** Next, we compare our method directly to the nonnegative low-rank (NLR) factorization. Figure 2 shows experimental results for GMM with $n = 100$, $\gamma = 0.8$, $\mu = 0.1$, and with varying $n$. Main observations: (i) Each Newton step is solved in $O(n)$ time, matching the theory in Section 3. (ii) A Newton step is about 25–100 times costlier than a single NLR update. This is to be expected because the Newton step solves several linear systems, while NLR performs only a single matrix-vector product. (iii) Our solver reaches the optimum in hundreds of iterations, whereas NLR needs tens of thousands. The orders-of-magnitude reduction in iterations more than offsets the costlier step, so wall-clock time drops by a factor of two to four.

**Comparison with prior Riemannian $K$-means methods.** We evaluate the clustering performance of the algorithm proposed by Carson et al. (2017) to solve the penalized formulation (4) on the CyTOF data. Figure 4 presents the performance of this first-order manifold method. Unfortunately, we were unable to identify a sequence of $\lambda$ that would produce acceptable clustering results. While increasing the penalty parameter $\lambda$ improves feasibility, it also degrades clustering performance. This highlights the difficulty in solving (4) using the method of Carson et al. (2017), as it struggles to balance strict constraint satisfaction with objective optimality in the $K$-means problem.

**Comparison with classical Riemannian algorithms.** As discussed in Section 3, Problem (13) can be solved with CG or RTR as the gradient, Hessian, and retraction are all available. Nevertheless, both CG and RTR perform poorly because the log-barrier induces an extremely ill-conditioned landscape. To illustrate this, we solve (13) on GMM data using PYMANOPT's implementation of CG and RTR. For CG, we were unable to tune the method to produce a meaningful solution, as its updates frequently lead to infeasible points. While RTR can converge to a solution comparable to ours, it requires significantly more iterations and time, as shown in Figure 5.

**Effect of hyperparameters.** Our method displays a sharp phase transition with respect to the regularization parameter $\mu$. Below a critical threshold, the algorithm consistently converges to

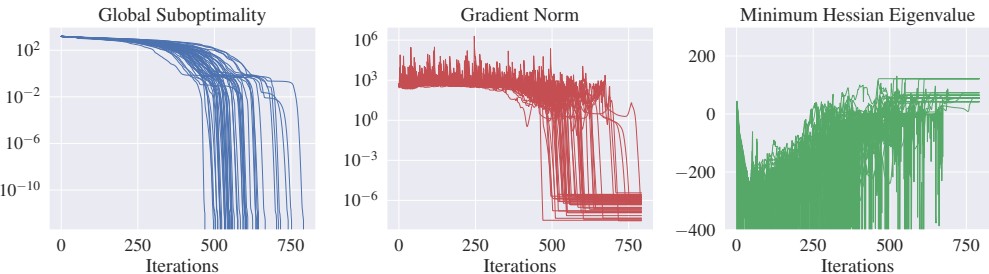

Figure 1: **Local convergence to second-order critical points yields global optimality.** In the GMM setting, where ground-truth partitions can be planted, we consistently observe local convergence to the global optimum, yielding zero clustering error. This provides strong numerical evidence that near-second-order critical points are near-globally optimal, as hypothesized in Assumption 1.

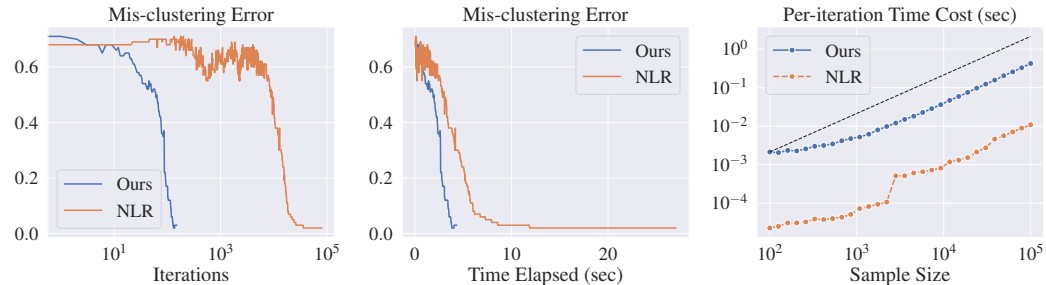

Figure 2: **Comparison with previous state-of-the-art NLR on GMM.** Our second-order method reaches optimality in 152 iterations, while NLR needs 80k. Even though each second-order iteration costs $\approx$ 25–100 NLR steps, the total runtime is still two to four times shorter. (Left and middle) clustering accuracy vs log iterations and linear time. (Right) per-iteration time vs sample size $n$.

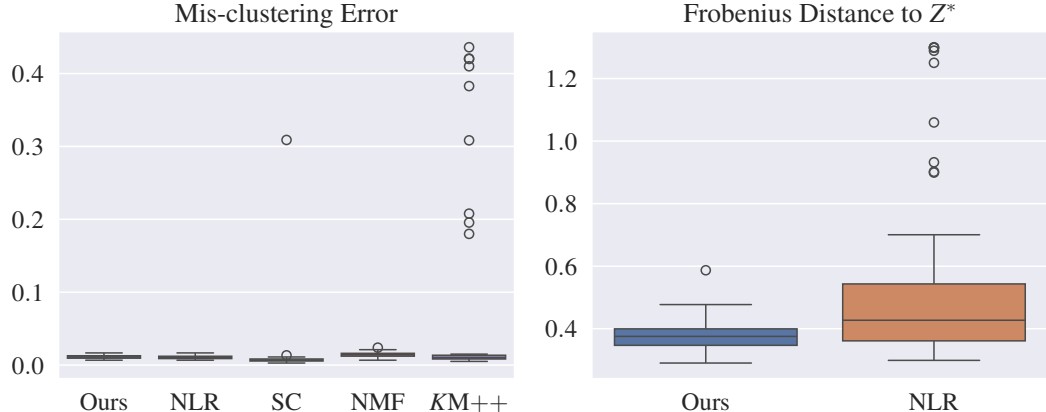

Figure 3: **Real-world benchmark on CyTOF data.** We compared our method to NLR, the previous state-of-the-art, as well as classical benchmarks SC, NMF, and $K$M++. Our method and NLR achieve the most consistently accurate clustering, with the smallest variance and the fewest outliers (left), but we outperform NLR in ground truth recovery (right).

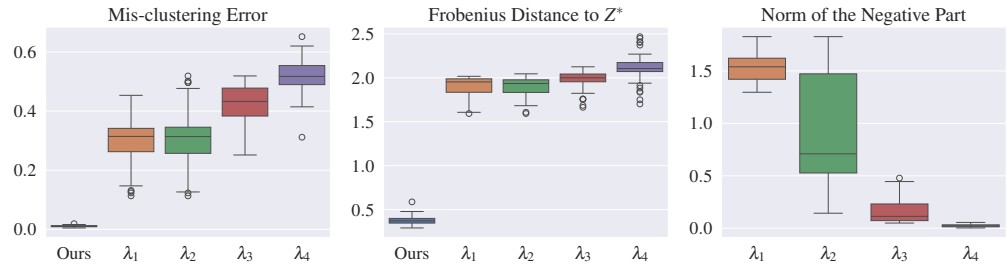

Figure 4: **Comparison with prior Riemannian $K$-means method of Carson et al. (2017) on real-world data.** Each run is warm-started from the previous and the penalty is stepped through $\lambda_i = 0, 10^4, 10^6, 10^7$. However: (Left) average mis-clustering exceeds 30%; (Middle) the recovery error $\|Z - Z^\star\|_F$ remains large; (Right) the infeasibility $\|U_-\|$ never vanishes. Our Riemannian method, shown for reference, enforces $U_- = 0$ by design and achieves near-zero error in both metrics.

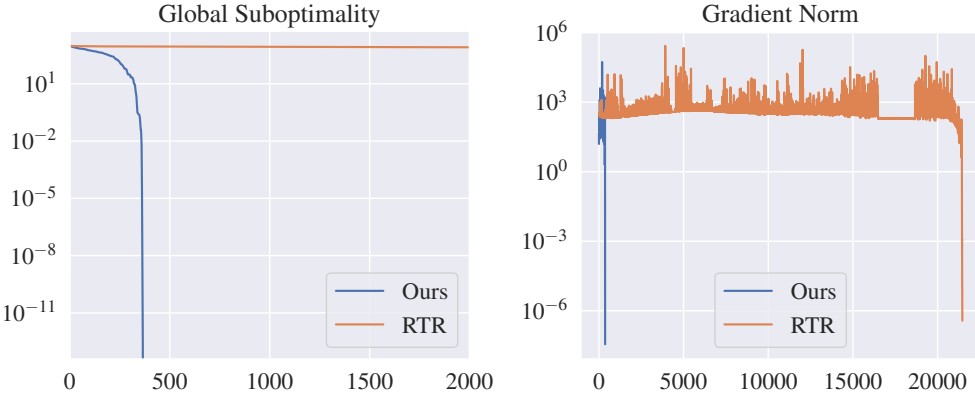

Figure 5: **Comparison with classical Riemannian Trust Region (RTR) on GMM.** Our method drives both loss and gradient norm to machine precision in around 360 iterations. In contrast, RTR stagnates for over 21k iterations due to the extreme ill-conditioning induced by the log penalty.

optimal solutions and remains robust to variations in $\mu$. However, once $\mu$ exceeds this threshold, the method fails to yield meaningful results. This behavior is illustrated in Figure 6 where we used GMM synthetic data with four clusters and separation $\gamma = 0.8$. Figure 6 also shows the clustering errors evaluated across different search ranks $r$. Clustering performance is largely insensitive to $r$ when $\mu$ is small, so the smallest feasible $r = K + 1$ is recommended in practice. However, when $\mu$ is large enough to trigger the phase transition, increasing $r$ can help the algorithm escape spurious regions. A brief discussion on the choice of hyperparameters is provided in Section G.

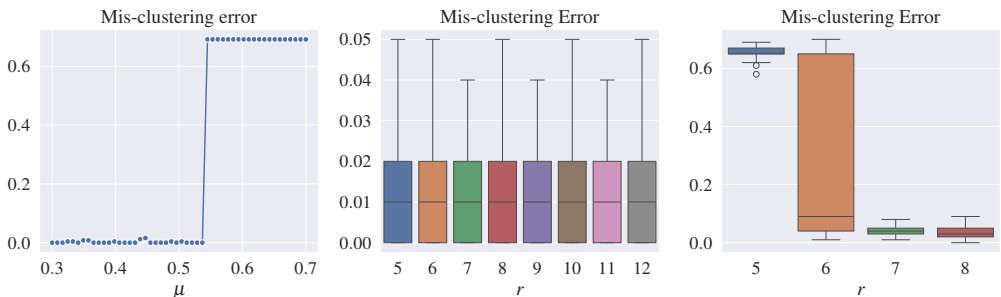

Figure 6: **Dependence on hyperparameters.** (Left) Error exhibits phase transition as penalty parameter $\mu$ gets too large. (Middle): Errors are insensitive to the search rank $r$, when $\mu$ is chosen appropriately. (Right): When $\mu$ is too large and the algorithm becomes trapped in local minima, increasing $r$ can lead to significantly better results.

**Robustness to mis-specified cluster number.** Figure 7 shows the performance of our method under mis-specified cluster numbers in the GMM setting with $K = 4, n = 1000$, and $\gamma = 0.8$; The mis-specified cluster number is set to be $K_{\mathrm{mis}} = 3$ and $K_{\mathrm{mis}} = 5$. In both setups, we observed that our method exhibits strong robustness in statistical accuracy for label recovery. We used normalized mutual information (NMI) to quantify the agreement between true and recovered labels, as the two sets of labels have different categories. Moreover, we observed that our method locally converges to the global optima in a similar manner as the $K = 4$ case (cf. Figure 1).

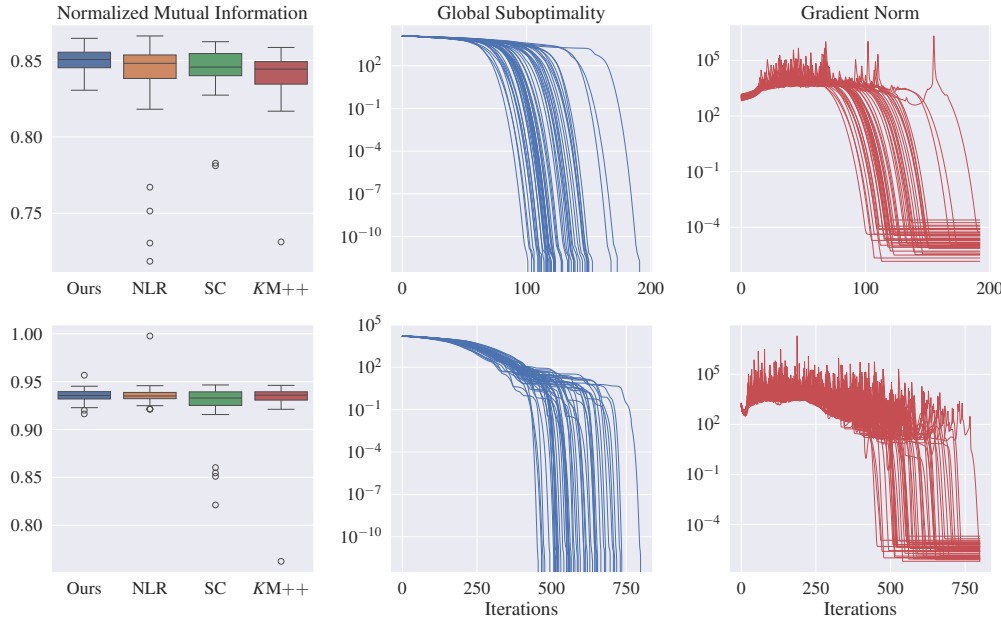

Figure 7: **Robustness to mis-specification.** Clustering performance and convergence behavior when the number of clusters is under-estimated (top) and over-estimated (bottom)

## ACKNOWLEDGMENTS

X. Chen was partially supported by NSF DMS-2413404 and a gift from the Simons Foundation. R. Zhang was partially supported by NSF CAREER Award ECCS-2047462 and ONR Award N00014-24-1-2671.

## REPRODUCIBILITY STATEMENT

The experimental setups are described in detail in Section 4 and Section G. While the provided code does not encompass every experiment reported in the paper, all of them can be readily reproduced based on the descriptions.

## LLM USAGE DISCLOSURE

A large language model (LLM) was used solely to polish the writing and improve clarity of expression. No part of the research ideation, discovery, or substantive content was generated by the LLM.

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

# A    RELATIONSHIP BETWEEN SDP FORMULATIONS OF $K$-MEANS CLUSTERING: STANDARD AND INTERIOR POINT VERSIONS (2)

The standard $K$-means clustering for data in $\mathbb{R}^d$ has two formulations in the literature: (i) *centroid-based* optimization

$$\min_{\beta_1,\ldots,\beta_K \in \mathbb{R}^d} \sum_{i=1}^n \min_{k \in [K]} \|X_i - \beta_k\|_2^2;$$

and (ii) *partition-based* optimization

$$\min_{\bigsqcup_{k=1}^K G_k = [n]} \sum_{k=1}^K \sum_{i \in G_k} \|X_i - \bar{X}_k\|_2^2,$$

where $\bar{X}_k = |G_k|^{-1} \sum_{j \in G_k} X_j$ is the empirical centroid of cluster $G_k$. Formulations (i) and (ii) are known to be equivalent, cf. Zhuang et al. (2022, Eqn. (1)) or Qian et al. (2022, Appx. A). Using the parallelogram law in Zhuang et al. (2022, Eqn. (5))

$$\sum_{i,j \in G_k} \|X_i - X_j\|_2^2 = 2|G_k| \sum_{i \in G_k} \|X_i - \bar{X}_k\|_2^2,$$

we may write the partition-based objective function as

$$\min_{\bigsqcup_{k=1}^K G_k = [n]} \sum_{k=1}^K \frac{1}{2|G_k|} \sum_{i,j \in G_k} \|X_i - X_j\|_2^2.$$

Next, expanding the pairwise squared Euclidean distance and dropping $\sum_{i=1}^n \|X_i\|_2^2$ (no longer depending on any partition $G_1,\ldots,G_K$), we arrive at (1), in the form of maximizing the total intra-cluster similarity in terms of the Gram matrix $\{\langle X_i, X_j \rangle\}_{i,j=1}^n$.

For general data without a likelihood derivation as in Chen & Yang (2021b), we can replace the $\mathbb{R}^d$-inner product with any (positive semidefinite) kernel $k \colon \mathcal{X} \times \mathcal{X} \to \mathbb{R}$ and consider the kernelized version of $K$-means clustering that involves data $X_1,\ldots,X_n$ only via their Gram matrix $\{k(X_i, X_j)\}_{i,j=1}^n$. Thus, our manifold formulation of this paper carries over to the general kernel method setting with possibly nonlinear boundary structure.

Next, we convexify the $K$-means problem (1) into an SDP. Note that each partition $G_1,\ldots,G_K$ via one-hot encoding is equivalent to an assignment matrix $H_{n \times K}$ (up to cluster relabel) where the latter is a binary matrix with exactly one non-zero entry in each row, i.e. $H_{ik} = 1$ if $i \in G_k$. With this reparameterization, one can write (1) as a mixed zero-one integer program:

$$\max_{H \in \{0,1\}^{n \times K}} \{ \langle XX^\top, HBH^\top \rangle : H\mathbf{1}_K = \mathbf{1}_n \}.$$

Now, applying the change of variables $Z_{n \times n} = HBH^\top$ and noting that the membership matrix $Z$ and assignment matrix $H$ are not one-to-one, we relax the $K$-means problem by preserving the key properties of $Z$ as the following constraints:

$$Z \succeq 0, \qquad \operatorname{tr}(Z) = K, \qquad Z\mathbf{1}_n = \mathbf{1}_n, \qquad Z \geq 0,$$

which no longer depend on the assignment matrix $H$. Then, we arrive at the standard SDP relaxation for $K$-means clustering, cf. Peng & Wei (2007, Eqn. (13)) or Chen & Yang (2021b, Eqn. (11)):

$$\max_{Z \in \mathbb{R}^{n \times n}} \left\{ \langle XX^\top, Z \rangle : Z \succeq 0, \operatorname{tr}(Z) = K, Z\mathbf{1}_n = \mathbf{1}_n, Z \geq 0 \right\}. \tag{15}$$

In practice, the elementwise nonnegativity constraint $Z \geq 0$ is almost always enforced by a logarithmic barrier. This means that we can make (15) more explicitly in the form

$$\max_{Z \in \mathbb{R}^{n \times n}} \left\{ \langle XX^\top, Z \rangle + \mu \sum_{i,j=1}^n \log(Z_{i,j})_+ : Z \succeq 0, \operatorname{tr}(Z) = K, Z\mathbf{1}_n = \mathbf{1}_n \right\}, \tag{16}$$

which is precisely how any practical interior-point solver would solve the original SDP in (15). The barrier cost is the actual objective used internally, with $\mu$ set to reflect the solver's target accuracy, cf. Boyd & Vandenberghe (2004, Chapter 11) or Nocedal & Wright (2006, Chapters 14 & 19). In other words, we take the standard SDP in (15), and make explicit the logarithmic penalty that is already implicit in how such an SDP is actually solved.

## B  TIGHTNESS OF PROPOSED FORMULATION

As shown in Carson et al. (2017), the combinatorial $K$-means problem (1) is exactly equivalent to

$$\max_{U \in \mathbb{R}^{n \times K}} \left\{ \langle XX^\top, UU^\top \rangle : U \geq 0, UU^\top \mathbf{1}_n = \mathbf{1}_n, U^\top U = I_K \right\}.$$

Formulation (3) replaces the hard constraint $U \geq 0$ with a log barrier, and then relax $U^\top U = I_K$ into $\text{tr}(UU^\top) = K$. Compare to (2), the containment $\{UU^T : U \geq 0\} \subset \{Z : Z \succeq 0, Z \geq 0\}$ is strict in general, so replacing $UU^\top$ by $Z$ yields a relaxation. Chen & Yang (2021b) showed that the resulting relaxation from (1) to (2) is exactly tight above the statistical separation threshold: its optimizer factors as $Z = UU^\top$ with $U \geq 0$, and the sparsity pattern of $U$ recovers the true cluster labels. Since (3) is tighter than (2), it must also be tight in this regime.

## C  INFORMATION-THEORETIC THRESHOLD FOR EXACT RECOVERY

The following theorem is a precise statement of the information-theoretic threshold of Chen & Yang (2021b).

**Theorem 3** (Average-case phase transition for exact recovery). *Let $\alpha > 0$ and*

$$\Theta_{\min} := \min_{1 \leq j < k \leq K} \|\mu_j - \mu_k\|$$

*be the minimum centroid separation. Suppose that data $X_1, \ldots, X_n$ are generated from the Gaussian mixture model (5) with equal cluster size $|G_1^*| = \cdots = |G_K^*| = m$. Then we have the following dichotomy.*

1. *If $K \leq \log n / \log \log(n)$ and $\Theta_{\min} \geq (1 + \alpha)\overline{\Theta}$, then there exist constants $C_1, C_2 > 0$ depending only on $\alpha$ such that, with probability at least $1 - C_1(\log n)^{-C_2}$, the SDP (2) as $\mu \to 0^+$ (cf. Appendix A for the equivalence of two SDP formulations) has a unique solution that exactly recovers the true partition $G_1^*, \ldots, G_K^*$.*

2. *If $K \leq \log n$ and $\Theta_{\min} \leq (1 - \alpha)\overline{\Theta}$, then there exists a constant $C_3 > 0$ depending only on $\alpha$ such that*

$$\inf_{\hat{G}_1, \ldots, \hat{G}_K} \sup_{\Xi(n, K, \Theta_{\min})} \mathbb{P}(\exists k : \hat{G}_k \neq G_k^*) \geq 1 - \frac{C_3 K}{n},$$

*where the infimum is taken over all possible estimators $(\hat{G}_1, \ldots, \hat{G}_K)$ for $(G_1^*, \ldots, G_K^*)$ and the parameter space is defined as*

$$\Xi(n, K, \Theta) := \left\{ (G_1, \ldots, G_K, \mu_1, \ldots, \mu_k) \, \middle| \, \begin{array}{l} \|\mu_j - \mu_k\| \geq \Theta, \; \forall j, k \in [K], j \neq k \\ (1 - \delta_n)m \leq |G_k| \leq (1 + \delta_n)m \end{array} \right\}$$

*with $\delta_n = C\sqrt{K \log(n)/n}$ for some large enough constant $C > 0$.*

## D  PROOFS

*Proof of Lemma 1.* Note that the membership matrix $Z$ associated to a partition $G_1, \ldots, G_K$ contains a diagonal principal submatrix of rank $K$. The lemma follows from Theorem 4 in Kalofolias & Gallopoulos (2012). □

*Proof of Lemma 2.* Let $N$ denote the null space basis of $A$, such that $AN = 0$ and $N^\top N = I$. Then, we have $\lambda_{\min} = \lambda_{\min}(N^\top H N)$ and $p(\lambda) = N\hat{p}(\lambda)$ where $(N^\top H N + \lambda I)\hat{p}(\lambda) = -N^\top g$. Then, $\lambda > -\lambda_{\min}$ implies that $N^\top H N + \lambda I \succ 0$, so $p(\lambda) = -N(N^\top H N + \lambda I)^{-1} N^\top g$ is always well-defined. Moreover, $\|p(\lambda)\| = \|\hat{p}(\lambda)\|$ is monotonously decreasing because all the eigenvalues of $N^\top H N + \lambda I$ are strictly positive and increasing with $\lambda$. □

*Proof of Lemma 5.* To prove the first statement, note that the implication

$$U \in \mathbb{R}_{\geq 0}^{n \times K} \wedge UU^\top \mathbf{1}_n = \mathbf{1}_n \wedge U^\top U = I_K \implies U \in \mathbb{R}_{\geq 0}^{n \times K} \wedge UU^\top \mathbf{1}_n = \mathbf{1}_n \wedge \|U\|_F^2 = K$$

is straightforward. To see the converse, note that $UU^\top$ is a (doubly) stochastic matrix, $\operatorname{tr}(UU^\top) = K$, and $\operatorname{rank}(UU^\top) = K$, thus all the $K$ eigenvalues of $UU^\top$ are 1, *i.e.* $U \in \operatorname{St}(n, K)$.

For the second statement, it is trivial that

$$U = \left[ \frac{1}{\sqrt{|G_1|}} \mathbf{1}_{G_1} \quad \cdots \quad \frac{1}{\sqrt{|G_K|}} \mathbf{1}_{G_K} \right] \wedge \bigsqcup_{k=1}^{K} G_k = [n]$$
$$\implies U \in \mathbb{R}_{\geq 0}^{n \times K} \wedge UU^\top \mathbf{1}_n = \mathbf{1}_n \wedge U^\top U = I_K.$$

The converse is also true. Let us denote $G_k := \operatorname{supp}(u_k)$. Observe that $G_i \cap G_j = \varnothing$ for all $i \neq j$ since $U \geq 0$ and $u_i^\top u_j = 0$ for all $i \neq j$. Then $UU^\top \mathbf{1}_n = \mathbf{1}_n$ implies $u_k u_k^\top \mathbf{1}_{G_k} = \mathbf{1}_{G_k}$ and $\bigsqcup_{k=1}^{K} G_k = [n]$. Since $\|u_k\| = 1$, we have that $u_k = \mathbf{1}_{G_k}/\sqrt{|G_k|}$.

Finally, we will prove the third statement. Let $U$ be a group assignment matrix as defined by (35). Note that for all $V \in T_U \mathcal{M} \cap \mathcal{C}_U$, $V$ must satisfy $(UV^\top + VU^\top)\mathbf{1}_n = \mathbf{0}_n$, $\langle U, V \rangle = 0$, and $v_{i,j} \geq 0, \forall u_{i,j} = 0$. Define $A : [n] \to [K]$ to be the group assigning function, i.e. $A(i) = \sum_{k=1}^{K} k\mathbb{I}\{u_{i,k} \neq 0\}$, then

$$UV^\top = \left[ \frac{1}{\sqrt{|G_{A(i)}|}} v_{j,A(i)} \right]$$
$$= \begin{bmatrix} \frac{1}{\sqrt{|G_{A(1)}|}} v_{1,A(1)} & \frac{1}{\sqrt{|G_{A(1)}|}} v_{2,A(1)} & \cdots & \frac{1}{\sqrt{|G_{A(1)}|}} v_{n,A(1)} \\ \frac{1}{\sqrt{|G_{A(2)}|}} v_{1,A(2)} & \ddots & & \vdots \\ \vdots & & \ddots & \vdots \\ \frac{1}{\sqrt{|G_{A(n)}|}} v_{1,A(n)} & \cdots & \cdots & \frac{1}{\sqrt{|G_{A(n)}|}} v_{n,A(n)} \end{bmatrix}.$$

Observe that

(i) $A(i) = A(j) \iff i \in G_{A(j)} \iff j \in G_{A(i)}$

(ii) $i \notin G_{A(j)} \iff (u_{i,A(j)} = 0 \wedge u_{j,A(i)} = 0) \implies (v_{i,A(j)} \geq 0 \wedge v_{j,A(i)} \geq 0)$

(iii) $\langle U, V \rangle = 0 \iff \forall j \in [n], \sum_{i \in G_{A(j)}} v_{i,A(j)} = 0$.

Denote $\boldsymbol{w} := (UV^\top + VU^\top)\mathbf{1}$, then

$$
\begin{aligned}
w_j &= \sum_{i \in [n]} \left( \frac{1}{\sqrt{|G_{A(j)}|}} v_{i,A(j)} + \frac{1}{\sqrt{|G_{A(i)}|}} v_{j,A(i)} \right) \\
&= \sum_{i \in G_{A(j)}} \left( \frac{1}{\sqrt{|G_{A(j)}|}} v_{i,A(j)} + \frac{1}{\sqrt{|G_{A(i)}|}} v_{j,A(i)} \right) \\
&\quad + \sum_{i \notin G_{A(j)}} \left( \frac{1}{\sqrt{|G_{A(j)}|}} v_{i,A(j)} + \frac{1}{\sqrt{|G_{A(i)}|}} v_{j,A(i)} \right).
\end{aligned}
$$

By (i) and (iii), we know that

$$
\sum_{i \in G_{A(j)}} \left( \frac{1}{\sqrt{|G_{A(j)}|}} v_{i,A(j)} + \frac{1}{\sqrt{|G_{A(i)}|}} v_{j,A(i)} \right) = \sqrt{|G_{A(j)}|} v_{j,A(j)}.
$$

By (ii), we know that

$$
\sum_{i \notin G_{A(j)}} \left( \frac{1}{\sqrt{|G_{A(j)}|}} v_{i,A(j)} + \frac{1}{\sqrt{|G_{A(i)}|}} v_{j,A(i)} \right) \geq 0.
$$

Thus we can write

$$
w_j = \sqrt{|G_{A(j)}|} v_{j,A(j)} + R_j \quad \text{for some } R_j \geq 0.
$$

Next, we use proof by contradiction. Suppose there exists $V \in \mathrm{T}_U \mathcal{M} \cap \mathcal{C}_U$ such that $V \neq 0$, then there must be both positive and negative entries in $V$ to satisfy $\langle U, V \rangle = 0$. This implies that there exists $j_1 \in [n]$ such that $v_{j_1, A(j_1)} < 0$ since $v_{j,A(j)}$'s are the only entries that can take negative value. To satisfy $\langle U, V \rangle = 0$, there must exist $j_2 \in G_{A(j_1)}$ such that $v_{j_2, A(j_1)} > 0$. Then for such $j_2$,

$$
w_{j_2} = \sqrt{|G_{A(j_2)}|} v_{j_2, A(j_2)} + R_{j_2} = \sqrt{|G_{A(j_2)}|} v_{j_2, A(j_1)} + R_{j_2} > 0.
$$

This contradicts $(UV^\top + VU^\top)\mathbf{1} = \mathbf{0}$. Therefore, $\mathrm{T}_U \mathcal{M} \cap \mathcal{C}_U = \{0\}$. $\qquad\square$

**Lemma 3.** *If $K \geq 2$, then the set $\mathcal{M}$ defined in (12) satisfies linear independence constraint qualification (LICQ) for all $U \in \mathbb{R}^{n \times r}$, and is therefore a smooth submanifold of $\mathbb{R}^{n \times r}$.*

*Proof of Lemma 3.* Following Boumal et al. (2020), the set $\mathcal{M} = \{U \in \mathbb{R}^{n \times r} : \langle A_i, UU^\top \rangle = b_i \text{ for all } i\}$ is a smooth submanifold of $\mathbb{R}^{n \times r}$ if LICQ holds for all $U \in \mathcal{M}$, i.e., that $\sum y_i A_i U = 0$ if and only $y = 0$. For the definition in (12), we can verify that

$$
\begin{aligned}
\left\| \sum y_i A_i U \right\|_F^2 &= \|(\mathbf{1}y^\top + y\mathbf{1}^\top + y_0 I)U\|_F^2 \\
&= \|(\mathbf{1}y^\top + y\mathbf{1}^\top)U\|_F^2 + 2\langle (\mathbf{1}y^\top + y\mathbf{1}^\top)U, y_0 U \rangle + y_0^2 \|U\|_F^2 \\
&= [ny^\top(I + UU^\top)y + 2(\mathbf{1}^\top y)^2] + 4y_0(\mathbf{1}^\top y) + y_0^2 K \\
&\geq 2[(\mathbf{1}^\top y)^2 + 2y_0(\mathbf{1}^\top y) + y_0^2] + n\|y\|^2 + (K-2)y_0^2 \\
&= 2(\mathbf{1}^\top y + y_0)^2 + n\|y\|^2 + (K-2)y_0^2 \\
&\geq 2(\mathbf{1}^\top y + y_0)^2 + n\|y\|^2.
\end{aligned}
$$

So if $\sum y_i A_i U = 0$, then from the last line we conclude that $\mathbf{1}^\top y + y_0 = 0$ and $y = 0$, so $(y, y_0) = 0$ as claimed. The third line above is because $\langle (\mathbf{1}y^\top + y\mathbf{1}^\top)U, U \rangle = 2\langle y, UU^\top \mathbf{1} \rangle = 2(y^\top \mathbf{1})$ and

$$
\begin{aligned}
\left\| (\mathbf{1}y^\top + y\mathbf{1}^\top)U \right\|_F^2 &= \mathrm{tr}[(\mathbf{1}y^\top + y\mathbf{1}^\top)UU^\top(\mathbf{1}y^\top + y\mathbf{1}^\top)] \\
&= \mathrm{tr}[\mathbf{1}y^\top UU^\top y\mathbf{1}^\top + y\mathbf{1}^\top UU^\top \mathbf{1}y^\top + 2 \cdot \mathbf{1}y^\top UU^\top \mathbf{1}y^\top] \\
&= \mathrm{tr}[\mathbf{1}y^\top UU^\top y\mathbf{1}^\top + y\mathbf{1}^\top \mathbf{1}y^\top + 2 \cdot \mathbf{1}y^\top \mathbf{1}y^\top] \\
&= ny^\top UUy + ny^\top y + 2(\mathbf{1}^\top y)^2.
\end{aligned}
$$

$\qquad\square$

**Lemma 4** (Interior point construction for $\mathcal{M}_r$). *Given a $K \in \mathbb{N}$, for any $r$ such that $r > K$, for large enough $n$, we have the following two cases:*
**Case 1***: $n \equiv 0 \pmod{r}$*
*Denote $q := n/r$, let $U_0 = (x - y)I + y\mathbf{11}^\top$, where*

$$x = \frac{1}{r}\Big(1 + \sqrt{(r-1)(K-1)}\Big), \qquad y = \frac{1}{r}\Big(\sqrt{r-1} - \sqrt{K-1}\Big).$$

*Then $U = (1/\sqrt{q})\mathbf{1}_q \otimes U_0$ is an interior point of $\mathcal{M}_r$.*
**Case 2***: $n \not\equiv 0 \pmod{r}$*
*Let us denote $q := \lfloor n/r \rfloor$ and $p := n \bmod r$. Construct the block matrix $B \in \mathbb{R}^{r \times r}$:*

$$B = \begin{bmatrix} B_{1,1} & B_{1,2} \\ B_{2,1} & B_{2,2} \end{bmatrix},$$

*where*

$$B_{1,1} = (x-y)I + y\mathbf{1}_p\mathbf{1}_p^\top, \qquad\qquad B_{1,2} = z\mathbf{1}_p\mathbf{1}_{r-p}^\top,$$
$$B_{2,2} = (w-z)I + z\mathbf{1}_{r-p}\mathbf{1}_{r-p}^\top, \qquad\qquad B_{2,1} = y\mathbf{1}_{r-p}\mathbf{1}_p^\top,$$

*The coefficients $x, y, z$ and $w$ depends on $n$, $K$ and $r$. They will be specified in the proof. Then*

$$U = [I_n \quad \mathbf{0}]\,(\mathbf{1}_q \otimes B)$$

*is an interior point of $\mathcal{M}_r$.*

*Proof of Lemma 4.* For a general large enough $n$, $n$ is either divisible or nondivisible by $r$. We present two different constructions of an interior point of $\mathcal{M}_r$ corresponding to the two cases.
**Case 1**: $n \equiv 0 \pmod{r}$
We first construct a $U_0 \in \mathbb{R}^{r \times r}$ such that $U_0 U_0^\top \mathbf{1}_r = \mathbf{1}_r$, and $\|U_0\|_F^2 = K$. Using the ansatz $U_0 = (x-y)I + y\mathbf{11}^\top$, where $x, y > 0$, we can find $x$ and $y$ by solving the system:

$$\begin{cases} x + (r-1)y = 1 & (U_0 U_0^\top \mathbf{1}_r = \mathbf{1}_r) \\ x^2 + (r-1)y^2 = K/r & (\|U_0\|_F^2 = K) \end{cases}.$$

The first equation gives $x = 1 - (r-1)y$. By substituting into the second equation, we obtain the following quadratic equation of $y$:

$$r(r-1)y^2 - 2(r-1)y + 1 - \frac{K}{r} = 0.$$

By the quadratic formula and $x, y > 0$, we have the following solution

$$\begin{cases} x = 1 - (r-1)y = \dfrac{1}{r}(1 + \sqrt{(r-1)(K-1)}), \\ y = \dfrac{r - 1 - \sqrt{(r-1)(K-1)}}{r(r-1)} = \dfrac{1}{r}(\sqrt{r-1} - \sqrt{K-1}). \end{cases}$$

Note that if $r = K$, we can still solve the quadratic equation, but without an all positive solution. Denote $q := n/r$, then $U = (1/\sqrt{q})\mathbf{1}_q \otimes U_0$ is an interior point of $\mathcal{M}_r$.
**Case 2**: $n \not\equiv 0 \pmod{r}$
Let us denote $q := \lfloor n/r \rfloor$ and $p := n \bmod r$. We consider the ansatz

$$U = [I_n \quad \mathbf{0}]\,(\mathbf{1}_q \otimes B) \quad \text{for some block matrix } B = \begin{bmatrix} B_{1,1} & B_{1,2} \\ B_{2,1} & B_{2,2} \end{bmatrix} \in \mathbb{R}^{r \times r},$$

where

$$B_{1,1} = (x-y)I + y\mathbf{1}_p\mathbf{1}_p^\top, \qquad\qquad B_{1,2} = z\mathbf{1}_p\mathbf{1}_{r-p}^\top,$$
$$B_{2,2} = (w-z)I + z\mathbf{1}_{r-p}\mathbf{1}_{r-p}^\top, \qquad\qquad B_{2,1} = y\mathbf{1}_{r-p}\mathbf{1}_p^\top.$$

Additional to the constraints $\|U\|_F^2 = K$, and $UU^\top = \mathbf{1}_n$, we assume that $U\mathbf{1}_r = c_r\mathbf{1}_n$, $U^\top\mathbf{1}_n = c_c\mathbf{1}_r$, and $c_rc_c = 1$ for some $c_r$ and $c_c$, which are sufficient for $UU^\top = \mathbf{1}_n$. Then we can find $x$, $y$, $z$, and $w$ by solving the system:

$$\begin{cases} x + (p-1)y + (r-p)z = py + w + (r-p-1)z, \\ (q+1)x + q(r-1)y + (p-1)y = qw + q(r-1)z + pz, \\ (py + w + (r-p-1)z)(qw + q(r-1)z + pz) = 1, \\ (q+1)p(x^2 + (p-1)y^2 + (r-p)z^2) + q(r-p)(py^2 + w^2 + (r-p-1)z^2) = K. \end{cases} \tag{17}$$

The four equations correspond to the following constraints, respectively: $U\mathbf{1}_r = c_r\mathbf{1}_n$, $U^\top\mathbf{1}_n = c_c\mathbf{1}_r$, $c_rc_c = 1$, and $\|U_0\|_F^2 = K$. From the first two equations, we can express $x$ and $y$ in terms of $z$ and $w$ (note that $n = qr + p$):

$$x = \left(1 - \frac{1}{n}\right)w + \frac{1}{n}z, \qquad y = \left(1 + \frac{1}{n}\right)z - \frac{1}{n}w. \tag{18}$$

By substituting (18) to the third and fourth equations of (17), we are left with a system of quadratic equations of two variables:

$$\begin{cases} a_1z^2 + a_2zw + a_3w^2 + c_1 = 0, \\ b_1z^2 + b_2zw + b_3w^2 + c_2 = 0, \end{cases} \tag{19}$$

where

$$\begin{aligned} & a_1 = nr - qr + p - p(2q+1)/n, && a_2 = 2p(1 + 2q - n)/n, \\ & a_3 = n - p(2q+1)/n, && b_1 = (r + p/n - 1)(n - q), \\ & b_2 = (1 - p/n)(n - q) + q(r + p/n - 1), && b_3 = q(1 - p/n), \\ & c_1 = -K, && c_2 = -1. \end{aligned}$$

Now our goal is to solve (19). By multiplying the first equation with $b_1$ and the second one with $a_1$ and subtraction, we can express $z$ in terms of $w$:

$$z = aw + \frac{b}{w} \quad \text{with } a = \frac{a_3b_1 - a_1b_3}{a_1b_2 - a_2b_1}, \ b = \frac{c_1b_1 - a_1c_2}{a_1b_2 - a_2b_1}. \tag{20}$$

Suppose that $w \neq 0$, we substitute (20) into the second equation and multiply by $w^2$. The result is a quartic equation:

$$(b_1a^2 + b_2a + b_3)w^4 + (2abb_1 + bb_2 + c_2)w^2 + b^2b_1 = 0. \tag{21}$$

By solving (21) with $z = aw + b/w > 0$, we obtain

$$w = \sqrt{\frac{-(2abb_1 + bb_2 + c_2) + \sqrt{(2abb_1 + bb_2 + c_2)^2 - 4(a^2b_1 + ab_2 + b3)b^2b_1}}{2(a^2b_1 + ab_2 + b_3)}}. \tag{22}$$

The proof is completed by combining (18), (20), and (22). $\qquad\square$

*Proof of Theorem 2.* For "$\supseteq$," if $U = \begin{bmatrix} \hat{\mathbf{1}}_n & V \end{bmatrix} Q$, then $UU^\top = n^{-1}\mathbf{1}_n\mathbf{1}_n^\top + VV^\top$ and hence $UU^\top\mathbf{1} = \mathbf{1}$ and $\text{tr}(UU^\top) = K$ respectively, because $\mathbf{1}_n^\top V = 0$ and $\text{tr}(VV^\top) = K-1$. For "$\subseteq$," let $U = P\Sigma Q$ denote the singular value decomposition with $P^\top P = QQ^\top = I_r$. Since $UU^\top\mathbf{1}_n = \mathbf{1}_n$, the decomposition can be chosen so that $P\Sigma e_1 = n^{-1/2}\mathbf{1}_n = \hat{\mathbf{1}}_n$. So if $V = P\Sigma[e_2, \ldots, e_r]$, then $\mathbf{1}_n^\top V = e_1^\top[e_2, \ldots, e_r] = 0$ and $\|V\|^2 = \|U\|^2 - \|P\Sigma e_1\|^2 = K-1$.

In the final part, we first construct the inner approximation $S$ of the tangent space

$$\begin{aligned} \mathrm{T}_{(V,Q)}\widetilde{\mathcal{M}} &= \{(\dot{V}, \dot{Q}) : \mathbf{1}_n^\top \dot{V} = 0, \langle V, \dot{V}\rangle = 0, Q\dot{Q}^\top + \dot{Q}Q^\top = 0\} \\ &\supseteq \left\{(\dot{V}, \dot{Q}) : \mathbf{1}_n^\top\dot{V} = 0, \langle V, \dot{V}\rangle = 0, \dot{Q}Q^\top = \begin{bmatrix} \mathbf{0} & -h^\top \\ h & \mathbf{0} \end{bmatrix}\right\} = S. \end{aligned}$$

We observe that

$$\begin{bmatrix} \hat{\mathbf{1}}_n & V \end{bmatrix} \dot{Q} = \begin{bmatrix} \hat{\mathbf{1}}_n & V \end{bmatrix} \begin{bmatrix} \mathbf{0} & -h^\top \\ h & \mathbf{0} \end{bmatrix} Q = \begin{bmatrix} Vh & \hat{\mathbf{1}}_nh^\top \end{bmatrix} Q$$

and therefore the Jacobian operator is injective for all $(\dot{V}, \dot{Q}) \in S$:

$$
\begin{aligned}
\left\| \mathrm{D}\,\varphi(V,Q)[\dot{V}, \dot{Q}] \right\|^2 &= \left\| \begin{bmatrix} \mathbf{0} & \dot{V} \end{bmatrix} Q + \begin{bmatrix} \hat{\mathbf{1}}_n & V \end{bmatrix} \dot{Q} \right\|^2 \\
&= \left\| \begin{bmatrix} \mathbf{0} & \dot{V} \end{bmatrix} \right\|^2 + \left\| \begin{bmatrix} Vh & \hat{\mathbf{1}}_n h^\top \end{bmatrix} \right\|^2 \geq \|\dot{V}\|^2 + \|h\|^2 \geq \frac{1}{\sqrt{2}} \left\| (\dot{V}, \dot{Q}) \right\|^2
\end{aligned}
$$

where we used the fact that $\mathbf{1}_n^\top \dot{V} = 0$. Hence, the Jacobian operator is surjective, as claimed:

$$
\dim\big(\mathrm{image}(\mathrm{D}\,\varphi(V,Q))\big) \geq \dim(S) = n(r-1) - 1 = \dim(\mathrm{T}_U\,\mathcal{M}).
$$

$\square$

# E  ADDITIONAL DETAILS ON RIEMANNIAN FORMULATION

## E.1  EQUIVALENCE BETWEEN RIEMANNIAN OPTIMIZATION AND SQP

Let us consider a constraint optimization problem,

$$
\begin{aligned}
\min_{u \in \mathbb{R}^n} \quad & f(u), \\
\text{s.t.} \quad & g_i(u) = 0 \ \forall i.
\end{aligned}
\tag{23}
$$

where LICQ:

$$
\sum_i y_i \nabla g_i(u) = 0 \iff y_i = 0 \ \forall i.
$$

holds for every $u$ in the constraint set. Then $\mathcal{M} \coloneqq \{u \in \mathbb{R}^n : g_i(u) = 0 \ \forall i\}$ is a smooth embedded manifold (Boumal, 2023), and we may employ Riemannian methods to solve (23).

The Riemannian method solves $\min_{u \in \mathcal{M}} f(u)$ by solving

$$
\min_{\dot{u} \in \mathrm{T}_u\,\mathcal{M}} f(u) + \langle \mathrm{grad}\, f(u), \dot{u} \rangle + \frac{1}{2} \langle \mathrm{Hess}\, f(u)[\dot{u}], \dot{u} \rangle + \frac{L}{6} \|\dot{u}\|^3
\tag{24}
$$

at each iteration. One can show that this is equivalent to the SQP method that solve (23) by minimizing the Lagrangian

$$
\begin{aligned}
\min_{\dot{u}} \quad & \mathcal{L}(u, y^{(u)}) + \langle \nabla_u \mathcal{L}(u, y^{(u)}), \dot{u} \rangle + \frac{1}{2} \langle \nabla_{uu}^2 \mathcal{L}(u, y^{(u)})[\dot{u}], \dot{u} \rangle + \frac{L}{6} \|\dot{u}\|^3, \\
\text{s.t.} \quad & \langle \nabla g_i(u), \dot{u} \rangle = 0 \ \forall i.
\end{aligned}
\tag{25}
$$

where

$$
\mathcal{L}(u, y^{(u)}) = f(u) + \sum_i y_i^{(u)} g_i(u) \quad \text{and} \quad y^{(u)} = \arg\min_y \| \nabla_u \mathcal{L}(u, y) \|.
$$

Hence, our contribution is to efficiently solve the SQP subproblem by exploiting a block-diagonal-plus-low-rank structure in the Hessian, and the fact that there are only $r + r(r+1)/2 \ll n$ constraints. We provide more details in Appendix E.2 and Section E.3.

To establish the equivalence, we first observe that the search space of (24) and (25) are the same. Indeed, the tangent space is $\mathrm{T}_u\,\mathcal{M} = \{\dot{u} : \langle \nabla g_i(u), \dot{u} \rangle = 0 \ \forall i\}$. Next, we write the expressions for the Riemannian gradient and Hessian (Boumal, 2023, Prop. 3.61 and Cor. 5.16):

$$
\mathrm{grad}\, f(u) \coloneqq \mathrm{Proj}_u\big(\nabla f(u)\big), \qquad \mathrm{Hess}\, f(u)[\dot{u}] \coloneqq \mathrm{Proj}_u\big(\mathrm{D}_u\, \mathrm{grad}\, f(u)[\dot{u}]\big),
\tag{26}
$$

where $\mathrm{Proj}_u(v) \coloneqq \arg\min_{\dot{u} \in \mathrm{T}_u\,\mathcal{M}} \|v - \dot{u}\|$, and $\mathrm{D}_u$ denotes the usual differential operator. We then obtain

$$
\mathrm{grad}\, f(u) \coloneqq \mathrm{Proj}_u\big(\nabla f(u)\big) = \nabla f(u) + \sum_i y_i^{(u)} \nabla g_i(u) = \nabla_u \mathcal{L}(u, y^{(u)}).
\tag{27}
$$

For the second equality, see Equation (7.75) in Boumal (2023).

For the second order terms, we have $\langle \mathrm{Hess}\, f(u)[\dot{u}], \dot{u} \rangle = \langle \nabla_{uu}^2 \mathcal{L}(u)[\dot{u}], \dot{u} \rangle$ for all $\dot{u}$ in $\mathrm{T}_u\,\mathcal{M}$ by the facts that

(i) $\operatorname{Hess} f(u)[\dot{u}] = \operatorname{Proj}_u\big(\nabla^2_{uu}\mathcal{L}(u, y^{(u)})[\dot{u}]\big)$.

(ii) $\langle \operatorname{Proj}_u(v), \dot{u} \rangle = \langle v, \dot{u} \rangle$.

We obtain fact (i) by (26) and (27):

$$
\begin{aligned}
\operatorname{Hess} f(u)[\dot{u}] &:= \operatorname{Proj}_u\big(\mathrm{D}\operatorname{grad} f(u)[\dot{u}]\big) \\
&= \operatorname{Proj}_u\big(\nabla^2 f(u)[\dot{u}] + \sum_i (\mathrm{D}_u\, y_i^{(u)})\nabla g_i(u)[\dot{u}] + \sum_i y_i^{(u)}\nabla^2 g_i(u)[\dot{u}]\big) \\
&= \operatorname{Proj}_u\big(\nabla^2 f(u)[\dot{u}] + \sum_i y_i^{(u)}\nabla^2 g_i(u)[\dot{u}]\big) \\
&= \operatorname{Proj}_u\big(\nabla^2_{uu}\mathcal{L}(u, y^{(u)})[\dot{u}]\big).
\end{aligned}
$$

Inside the projection operator, the term $\sum_i (Dy_i^{(u)})\nabla g_i(u)$ vanishes because $\mathrm{T}_u\mathcal{M}$ is a linear subspace, and $(\mathrm{T}_u\mathcal{M})^{\perp} = \operatorname{span}\big(\nabla g_i(u)\big)$. Fact (ii) is due to that the projection operator is self-adjoint and that the projection of any tangent vector is itself. For more on the connection between SQP and Riemannian Newton method, we refer to Absil et al. (2009); Mishra & Sepulchre (2016).

### E.2  GENERAL FORM OF THE RIEMANNIAN GRADIENT AND HESSIAN

Throughout this section, we use a bar over a function defined on the manifold $\mathcal{M}$ to denote its smooth extension defined on a neighborhood of $\mathcal{M}$ so that the Euclidean gradient and Hessian can be defined on $\mathcal{M}$. Namely, for $f: \mathcal{M} \to \mathbb{R}^m$, we use $\bar{f}: N(\mathcal{M}) \to \mathbb{R}^m$ to denote the smooth extension. The notations $\operatorname{grad} f$ and $\operatorname{Hess} f$ denote the Riemmanian gradient and Hessian; $\nabla\bar{f}$ and $\nabla^2\bar{f}$ denote the Euclidean gradient and Hessian.

For manifolds that can be defined by

$$
\min_{U \in \mathcal{M}} f(U), \qquad \mathcal{M} = \{U \in \mathbb{R}^{n \times r} : \mathcal{A}(UU^{\top}) + \mathcal{B}(U) = c\},
$$

where $\mathcal{A}: \mathbb{R}^{n \times n} \to \mathbb{R}^m$ and $\mathcal{B}: \mathbb{R}^{n \times r} \to \mathbb{R}^m$ are linear operators, and $c \in \mathbb{R}^m$, its tangent space can be written as:

$$
\mathrm{T}_U\mathcal{M} = \{\dot{U} \in \mathbb{R}^{n \times r} : \mathcal{A}(\dot{U}U^{\top} + U\dot{U}^{\top}) + \mathcal{B}(\dot{U}) = 0\}.
$$

We call the function $\mathcal{A}(UU^{\top}) + \mathcal{B}(U) = c$ as the *defining function* of $\mathcal{M}$. Let us denote

$$
L(\dot{U}) := \mathcal{A}(\dot{U}U^{\top} + U\dot{U}^{\top}) + \mathcal{B}(\dot{U}). \tag{28}
$$

Immediately, we see that $\mathrm{T}_U\mathcal{M} = \ker(L)$, and the adjoint operator

$$
L^*(y) = 2\,\mathcal{A}^{\top}(y)U + \mathcal{B}^{\top}(y). \tag{29}
$$

The projection operator onto the tangent space is defined to be

$$
\operatorname{Proj}_U(W) := \arg\min_{\dot{U} \in \mathrm{T}_U\mathcal{M}} \|W - \dot{U}\|.
$$

We also know that $\ker(L)^{\perp} = \operatorname{image}(L^*)$. Thus, by the orthogonal projection theorem, we may see that

$$
\operatorname{Proj}_U(W) = W - W^{\perp} = W - L^*(\tilde{y}), \tag{30}
$$

where $\tilde{y} = \arg\min_y \|W - L^*(y)\|$ is the solution to the linear system $W - L^*(y) \in \ker(L) = \mathrm{T}_U\mathcal{M}$, and both $\operatorname{Proj}_U(W)$ and $\tilde{y}$ are unique.

Consequently, the Riemannian gradient and the Hessian matrix-vector product have the following form:

$$
\begin{cases}
\operatorname{grad} f(U) := \operatorname{Proj}_U(\nabla\bar{f}(U)) = \nabla\bar{f}(U) + 2[\mathcal{A}^{\top}(y_U)]U + \mathcal{B}^{\top}(y_U), \\
\operatorname{Hess} f(U)[\dot{U}] := \operatorname{Proj}_U\big(\mathrm{D}\operatorname{grad} f(U)[\dot{U}]\big) = \operatorname{Proj}_U\big(\nabla^2\bar{f}(U)[\dot{U}] + 2[\mathcal{A}^{\top}(y_U)]\dot{U}\big),
\end{cases} \tag{31}
$$

where $y_U = -\tilde{y}$ in (30) is the unique Lagrange multipliers

$$
y_U = \arg\min_{y \in \mathbb{R}^m} \left\|\nabla f(U) + 2[\mathcal{A}^{\top}(y)]U + \mathcal{B}^{\top}(y)\right\|. \tag{32}
$$

For a detailed proof, see Boumal et al. (2020).

### E.3 EFFICIENT COMPUTATION OF THE RIEMANNIAN GRADIENT AND HESSIAN

We first write down the Euclidean gradient and Hessian for our objective function, and then explain how to compute the Riemannian counterparts efficiently. Specifically, we show that $y_U$ can be computed in $O(nr + r^3)$ time.

We decompose the objective function as $f = f_1 + \mu f_2$, where

$$f_1(V, Q) := \langle C, VV^\top \rangle, \qquad f_2(V, Q) := -\sum_{i,j} \log \varphi_{i,j}(V, Q).$$

For $f_1$, the Euclidean gradients are $\nabla_V f_1(V, Q) = 2CV$ and $\nabla_Q f_1(V, Q) = 0$, respectively. For $f_2$, the gradients are

$$\nabla_V f_2(V, Q) = -U_{(-1)}\hat{Q}^\top, \qquad \nabla_Q f_2(V, Q) = -\hat{V}^\top U_{(-1)},$$

where $U := \varphi(V, Q)$, and $U_{(-1)}$ is the element-wise reciprocal, i.e., $[U_{(-1)}]_{i,j} = U_{i,j}^{-1}$. For convenience, we also define $\hat{Q} := DQ$ with $D := [\mathbf{0}_{r-1} \quad I_{r-1}]$. Note that $\hat{Q}$ is simply $Q$ without the first row. Putting these together, we obtain

$$G_V := \nabla_V f(V, Q) = 2CV - \mu U_{(-1)}\hat{Q}^\top, \qquad G_Q := \nabla_Q f(V, Q) = -\mu \hat{V}^\top U_{(-1)}.$$

The Euclidean Hessians can be given in vectorized form as

$$\nabla^2 f_1 = \begin{bmatrix} 2I_{r-1} \otimes C & \mathbf{0} \\ \mathbf{0} & \mathbf{0} \end{bmatrix}, \qquad \nabla^2 f_2 = \begin{bmatrix} J_V^\top \\ J_Q^\top \end{bmatrix} \text{dvec}(U_{(-2)}) \begin{bmatrix} J_V & J_Q \end{bmatrix} - \begin{bmatrix} \mathbf{0} & H_{VQ} \\ H_{QV} & \mathbf{0} \end{bmatrix},$$

where

$$J_V = \hat{Q}^\top \otimes I_n, \qquad J_Q = I_r \otimes QU^\top,$$

and

$$H_{VQ} = (D \otimes U_{(-1)})K^{(r,r)} = H_{QV}^\top,$$

with $U_{(-2)}$ being the element-wise square of $U_{(-1)}$, $\text{dvec}(\cdot) := \text{diag}[\text{vec}(\cdot)]$, and $K^{(n,r-1)}, K^{(r,r)}$ denoting the commutation matrices (Magnus & Neudecker, 2019, Sec. 3.7). We can then compute the Riemannian gradient and Hessian-vector-product according to (31) and (32). In the remainder of this section, we show how to efficiently solve (32).

The manifold we consider, $\widetilde{\mathcal{M}} := \mathcal{V} \times \mathcal{O}_r$, can be written as

$$\widetilde{\mathcal{M}} = \left\{ (V, Q) \in \mathbb{R}^{n \times (r-1)} \times \mathbb{R}^{r \times r} : \mathbf{1}_n^\top V = \mathbf{0}, \text{tr}(VV^\top) = K - 1, \text{svec}(QQ^\top - I_r) = \mathbf{0} \right\},$$

where svec denotes the symmetric vectorization (Klerk, 2006, Appx. E). Note that there are no cross terms ($VQ^\top$ or $QV^\top$) in the defining functions of $\mathcal{M}$. Thus, we can treat the defining functions with respect to $V$ and $Q$ separately. The corresponding terms are

$$\mathcal{A}_V(VV^\top) := \begin{bmatrix} \mathbf{0}_{r-1} \\ \text{tr}(VV^\top) \end{bmatrix}, \qquad \mathcal{B}_V(V) := \begin{bmatrix} \mathbf{1}_n^\top V \\ 0 \end{bmatrix}, \qquad \mathcal{A}_Q(QQ^\top) := \text{svec}(QQ^\top)$$

and

$$c_V := \begin{bmatrix} \mathbf{0}_{r-1} \\ K - 1 \end{bmatrix} \quad \text{and} \quad c_Q := \text{svec}(I_r).$$

Mimicking (28) and (29), we use the notation $L_V$, $L_Q$, $L_V^*$, and $L_Q^*$ respectively. For any $(y_1, y_2) \in \mathbb{R}^{r-1} \times \mathbb{R}$ and $y_3 \in \mathbb{R}^{r(r+1)/2}$, we have

$$L_V^*(y_1, y_2) = \mathbf{1}_n y_1^\top + 2y_2 V \quad \text{and} \quad L_Q^*(y_3) = 2 \text{smat}(y_3)Q,$$

where smat is the inverse of svec, that is, $\text{smat}(\text{svec}(M)) = M$ for all symmetric matrices $M$. By solving the linear systems

$$G_V - L_V^*(y_1, y_2) \in \ker(L_V) \quad \text{and} \quad G_Q - L_Q^*(y_3) \in \ker(L_Q),$$

we obtain the following closed form solutions:

$$\tilde{y}_1 = \frac{1}{n} G_V^\top \mathbf{1}_n, \qquad \tilde{y}_2 = \frac{\langle G_V, V \rangle}{2(K-1)}, \qquad \tilde{y}_3 = \frac{1}{4} \operatorname{svec}(G_Q Q^\top + Q G_Q^\top). \tag{33}$$

The computation of $\tilde{y}_1$, $\tilde{y}_2$, and $\tilde{y}_3$ in total requires $O(nr + r^3)$ time. Therefore, we can compute $y_U = -(\tilde{y}_1, \tilde{y}_2, \tilde{y}_3)$ with the same cost. Given a Euclidean gradient and a Euclidean Hessian-vector product, we may write out explicitly the Riemannian gradient:

$$\operatorname{grad} f(V, Q) = \left[ \left( I_n - \frac{1}{n} \mathbf{1}_n \mathbf{1}_n^\top \right) G_V - \frac{\langle G_V, V \rangle}{K-1} V \quad \frac{G_Q Q^\top - Q G_Q^\top}{2} Q \right] \tag{34}$$

and the Riemannian Hessian-vector product:

$$\operatorname{Hess} f(V, Q)[\dot{V}, \dot{Q}] = \begin{bmatrix} \operatorname{Proj}_V \left( \nabla^2 f(V, Q)[\dot{V}, \dot{Q}]_V - \frac{\langle G_V, V \rangle}{K-1} \dot{V} \right) \\ \operatorname{Proj}_Q \left( \nabla^2 f(V, Q)[\dot{V}, \dot{Q}]_Q - \frac{G_Q Q^\top + Q G_Q^\top}{2} \dot{Q} \right) \end{bmatrix}^\top .$$

### E.4 FEASIBLE INITIAL POINT

In this section, we show that $r > K$ is necessary and sufficient for the existence of an interior point of $\mathcal{M}$. The following Lemma 5 shows the necessity of $r > K$. When $r = K$, the structure of the unique $U \in \mathbb{R}_+^{n \times K}$ in Lemma 1 can be explicitly written as

$$U = \left[ \frac{1}{\sqrt{|G_1|}} \mathbf{1}_{G_1}, \frac{1}{\sqrt{|G_2|}} \mathbf{1}_{G_2}, \ldots, \frac{1}{\sqrt{|G_K|}} \mathbf{1}_{G_K} \right], \tag{35}$$

where $\mathbf{1}_{G_k} \in \{0, 1\}^n$ denotes the binary vector with its support being $G_k$.

**Lemma 5** (Isolated feasibility when $r = K$). *Let $\mathcal{M}_+ = \mathcal{M} \cap \mathbb{R}_+^{n \times K}$ and $\mathcal{M}'_+ = \mathcal{M}' \cap \mathbb{R}_+^{n \times K}$, where $\mathbb{R}_+^{n \times K} = \{U \in \mathbb{R}^{n \times K} : U \geq 0\}$. Then, we have: (i) $\mathcal{M}_+ = \mathcal{M}'_+$; (ii) $U \in \mathcal{M}_+$ if and only if $U$ is a group assignment matrix defined in (35); (iii) if $U$ is a group assignment matrix, then the intersection of the tangent space $\mathrm{T}_U \mathcal{M}$ and the cone $\mathcal{C}_U := \{V \in \mathbb{R}^{n \times K} : v_{ij} \geq 0, \forall u_{ij} = 0\}$ is trivial, i.e., $\mathrm{T}_U \mathcal{M} \cap \mathcal{C}_U = \{0\}$.*

In Lemma 4, we moreover provide a complete analytical construction for an interior point of $\mathcal{M}$ when $r > K$. Here, we present the construction when $n = qr$ is an integer multiple of $r$. Let $U_0 = (x - y)I + y \mathbf{1}_n \mathbf{1}_n^\top$, where $x = r^{-1}(1 + \sqrt{(r-1)(K-1)})$ and $y = r^{-1}(\sqrt{r-1} - \sqrt{K-1})$.

Then $U = \hat{\mathbf{1}}_q \otimes U_0$ is an interior point of $\mathcal{M}$. Next, we show how to compute the pair $V$, $Q$ corresponding to the interior point $U$ by SVD. For a given $U \in \mathcal{M}_r$, let $U = P_U \Sigma Q_U^\top$ be the SVD of $U$. We can find $(V, Q)$ such that $U = \hat{V} Q$ by

$$V = \operatorname{sgn}(P_{U\,(1,1)}) \cdot P_U \Sigma [e_2, \ldots, e_r] \quad \text{and} \quad Q = \operatorname{sgn}(P_{U\,(1,1)}) \cdot Q_U^\top.$$

### E.5 LIPSCHITZ CONTINUITY OF PENALTY

To apply the guarantees in Section 2.3, we need to take care of the logarithmic penalty in (13) since it does not have Lipschitz gradients nor Hessians over its whole domain. The standard workaround, widely used in the analysis of nonlinear interior-point methods, is to observe that all iterates $U_k = \varphi(V_k, Q_k)$ remain strictly feasible. Consequently, the penalty could be modified by a Huber-style smoothing, where $\delta = \min_{i,j,k}(U_k)_{i,j} > 0$:

$$r(x) = \begin{cases} \log x & x \geq \delta \\ \log \delta + \dfrac{(x - \delta)}{\delta} - \dfrac{(x - \delta)^2}{2\delta^2} + \dfrac{(x - \delta)^3}{2\delta^3} & x < \delta \end{cases}$$

The function $r(x)$ is both concave and has Lipschitz Hessians. Therefore, the guarantees in Section 2.3 apply. The smoothing is only needed for theoretical purposes. In practice, we apply the Riemannian algorithms directly to $\log x$, and not to $r(x)$. Since we have assumed that all queries satisfy $x \geq \delta$, the actual behavior remains consistent with the smoothed model.

## F   EFFICIENT IMPLEMENTATION AND COST OF BISECTION SEARCH

To implement the proposed method, we vectorize the input as $u = (v, q)$, with $v := \text{vec}(V^\top)$ and $q := \text{vec}(Q)$. Since $C = -XX^\top$, we rewrite the cost function from Section E.3 as

$$f(u) = -\|X^\top V\|^2 - \mu \mathbf{1}_n^\top \log\big(\varphi(V, Q)\big)\mathbf{1}_n,$$

and define the constraint functions

$$g_1(u) = \mathbf{1}_n^\top V, \qquad g_2(u) = \|V\|^2 - (K - 1), \qquad g_3(u) = \text{svec}(QQ^\top - I_r).$$

The Jacobian $J$ and Hessian $H$ are computed analytically, as in Section E.3, in order to exploit their sparsity.

Since we have restricted ourselves to optimizing over $p \in \mathrm{T}_u \widetilde{\mathcal{M}}_r$, we may replace the Riemannian Hessian $H$ with $\tilde{H} := \nabla^2_{uu} \mathcal{L}(u, y_u)$. The key observation is that $\tilde{H}$ admits a block-diagonal-plus-low-rank structure.

For convenience, we list some of the derivatives in this section. The (Euclidean) Jacobian of the constraints can be written in block form as

$$J = \begin{bmatrix} J_{1v} & \mathbf{0} \\ J_{2v} & \mathbf{0} \\ \mathbf{0} & J_{3q} \end{bmatrix},$$

where

$$J_{1v} = (I_{r-1} \otimes \mathbf{1}_n^\top)K^{(r-1,n)}, \qquad J_{2v} = 2v, \qquad J_{3q} = S_r(I_{r^2} + K_{r,r})(Q \otimes I_r),$$

where $S_r \in \mathbb{R}^{r(r+1)/2 \times r^2}$ is the matrix satisfying $S \text{vec}(Q) = \text{svec}(Q)$.

Computing the second-order derivatives of $g_1, g_2$ is straightforward, since $J_{1v}$ is constant in $V$ and $J_{2v}$ is linear. To compute the second-order derivatives of $g_3$, since we only need to compute $\sum_{j \in [r(r+1)/2]} \tilde{y}_{3,j} \nabla^2 g_{3,j}(q)$, we note that for any $y \in \mathbb{R}^{r(r+1)/2}$,

$$J_{3q}^\top y = 2\,\text{vec}(\text{smat}(y)Q).$$

Consequently, we have for any $y \in \mathbb{R}^{r(r+1)/2}$,

$$\sum_{j=1}^{r(r+1)/2} y_j \nabla^2 g_{3,j} = 2I_r \otimes \text{smat}(y).$$

Collecting results, we have

$$\tilde{H} = \begin{pmatrix} H_{vv} - BB^\top & H_{vq} \\ H_{qv} & H_{qq} \end{pmatrix},$$

with

$$B = \sqrt{2}(X \otimes I_{r-1}) \tag{36}$$

$$H_{vv} = \mu(I_n \otimes \hat{Q})\,\text{dvec}\Big(U_{(-2)}^\top\Big)(I_n \otimes \hat{Q}^\top) + 2\tilde{y}_2 I, \tag{37}$$

$$H_{qq} = \mu(I_r \otimes \hat{V}^\top)\,\text{dvec}\big(U_{(-2)}\big)(I_r \otimes \hat{V}) + 2I_r \otimes \text{smat}(\tilde{y}_3), \tag{38}$$

and

$$H_{vq} = -\mu(U_{(-1)} \otimes D) + \mu(I_n \otimes \hat{Q})K^{(n,r)}\,\text{dvec}\big(U_{(-2)}\big)(I_r \otimes \hat{V}) = H_{qv}^\top, \tag{39}$$

where $\tilde{y}_2, \tilde{y}_3$ follow (33).

To solve the saddle point problem arising from subproblem 14 via bisection search, we solve the linear system

$$\left[\begin{array}{cc|cc} H_{vv} + \lambda I - BB^\top & & H_{vq} & J_v^\top \\ H_{qv} & & H_{qq} + \lambda I & J_q^\top \\ J_v & & J_q & \mathbf{0} \end{array}\right] \begin{bmatrix} \dot{v} \\ \dot{q} \\ z_0 \end{bmatrix} = \begin{bmatrix} -g_v \\ -g_q \\ \mathbf{0} \end{bmatrix}.$$

Repartitioning along the lines yields:

$$\left[\begin{array}{c|c} K_{11} & K_{12} \\ \hline K_{21} & K_{22} \end{array}\right]\left[\begin{array}{c} x_1 \\ \hline x_2 \end{array}\right] = \left[\begin{array}{c} b_1 \\ \hline b_2 \end{array}\right]$$

with $x_1, b_1 \in \mathbb{R}^{n(r-1)}$ and $x_2, b_2 \in \mathbb{R}^{r^2+m}$. In particular, we observe that the block $K_{11}$ has the form $K_{11} = D_{11} - BB^\top$, where $D_{11} = H_{vv} + \lambda I$ is block-diagonal, with $n$ blocks of $r-1$, and $B$ has at most $dr$ columns. Therefore, we instead solve

$$\left[\begin{array}{c|cc} D_{11} & K_{12} & B \\ \hline K_{21} & K_{22} & \mathbf{0} \\ B^\top & \mathbf{0} & I \end{array}\right]\left[\begin{array}{c} \dot{v} \\ \hline x_2 \\ z_1 \end{array}\right] = \left[\begin{array}{c} b_1 \\ \hline b_2 \\ \mathbf{0} \end{array}\right].$$

First, it costs $n(r-1)^3 = O(nr^3)$ time to invert $D_{11}$. Afterwards, forming and solving the size $m + r^2 + rd$ Schur complement problem:

$$\left(L_{22} - L_{12}^\top D_{11}^{-1} L_{12}\right)\begin{bmatrix} x_2 \\ z_1 \end{bmatrix} = \begin{bmatrix} b_2 \\ \mathbf{0} \end{bmatrix} - L_{12}^\top D_{11}^{-1} b_1, \tag{40}$$

where

$$L_{12} := \begin{bmatrix} K_{21} & B \end{bmatrix}, \qquad L_{22} := \begin{bmatrix} K_{22} & \mathbf{0} \\ \mathbf{0} & I \end{bmatrix},$$

cost $O(nr^3(d+r) + r^6 + r^3 d^3)$ time. In the end, we substitute to recover

$$\dot{v} = D_{11}^{-1}\left(b_1 - L_{12}\begin{bmatrix} x_2 \\ z_1 \end{bmatrix}\right)$$

in $O(nr^3(d+r))$ time, and apply retractions to $\dot{v}$ and $\dot{q} = [I_{r^2} \quad 0]x_2$. In total, it takes $O(nr^3(d+r) + r^6 + r^3 d^3)$ time to solve the system, which is indeed $n \cdot \text{poly}(r, d)$. Putting pieces together, a pseudo-code of our Riemannian method is shown in Algorithm 1.

## G ADDITIONAL NUMERICAL RESULTS

We collect additional numerical results in this section.

**Dataset visualization.** Figure 8 and Figure 9 display the first two principal components of the GMM dataset and CyTOF dataset, respectively.

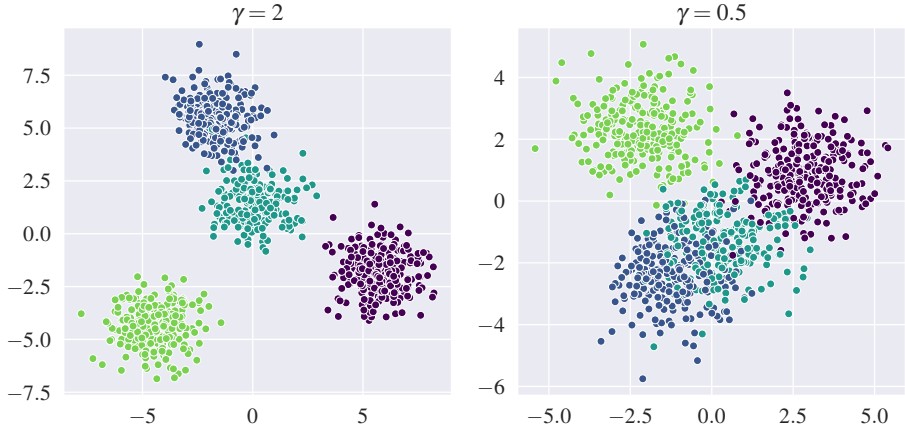

Figure 8: Visualizing the effect of the separation parameter in GMMs. As $\gamma$ decreases, the clusters become increasingly difficult to distinguish.

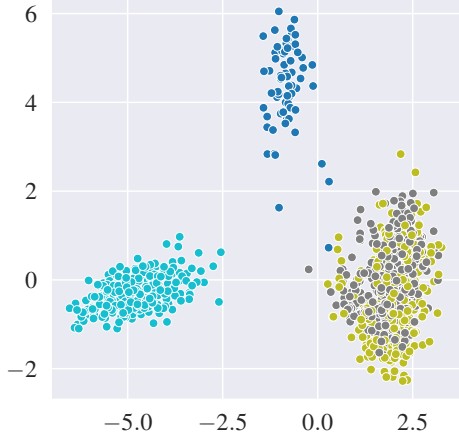

Figure 9: Visualization of the CyTOF dataset. Two clusters exhibit significant overlap, implying the difficulty of clustering.

**Benchmark on CIFAR10.** Figure 10 shows the performance of our method on the CIFAR10 image classification dataset. The original $32 \times 32$ color images were processed using a pre-trained Vision Transformer (ViT-B-16) (Dosovitskiy et al., 2021) to extract 768-dimensional features, which were then reduced to $d = 50$ using PCA. We use a pre-trained model to avoid breaching the unsupervised setting. From the dataset, 25000 images across five classes were selected; In each trial, we draw a subsample of 1,000 images and perform clustering. We repeated this procedure 50 times.

**Generalization to kernelized $K$-mean.** Since the data enter the problem through the Gram matrix $C = -X X^\top$ in Equation (3), extending the implementation to the kernel $K$-means is straightforward.

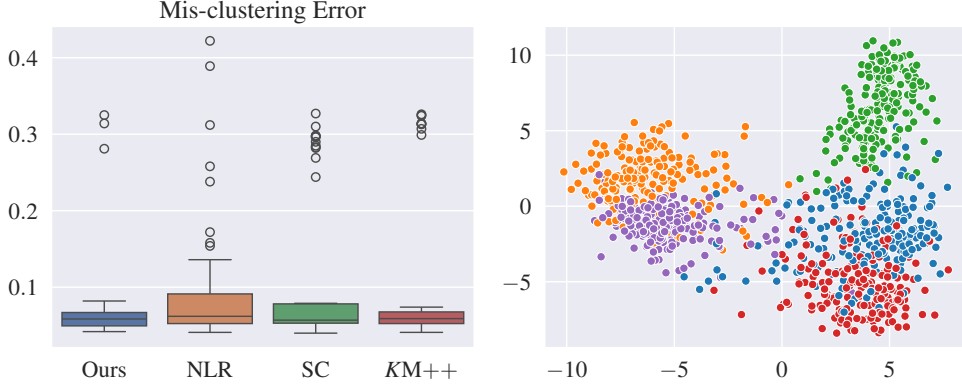

Figure 10: Real-world benchmark on the CIFAR10 data. (Left): comparison to other state-of-the-art methods. (Right): first two principal components of the image embeddings.

We adapted our implementation to accept a kernel matrix as input and conducted experiments using the RBF kernel on two toy datasets. The results are shown in Figure 11.

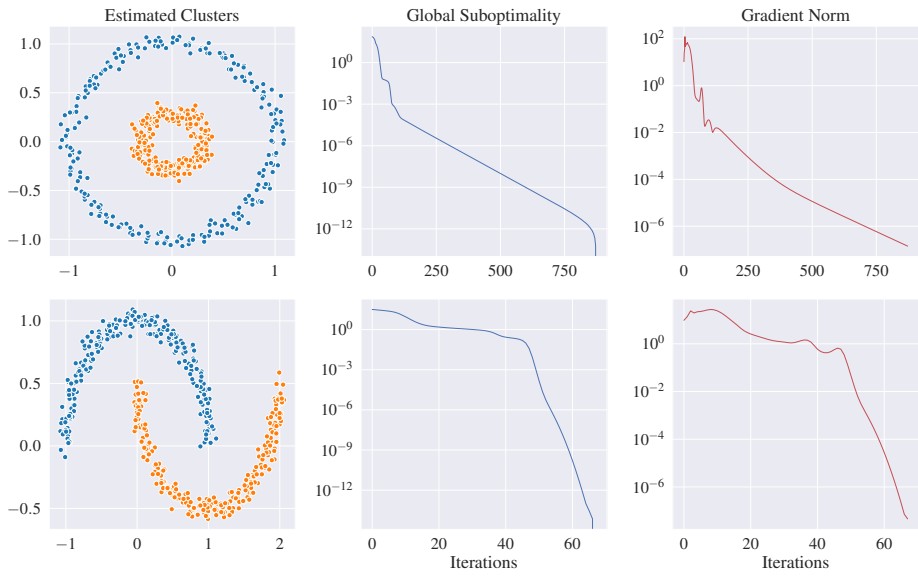

Figure 11: Clustering with the RBF kernel. Demonstrated on the Circles and Moon datasets from `scikit-learn`. The kernel bandwidths are 3 and 15, respectively.

Nevertheless, without access to the factored matrix, we can no longer perform the sparse partition described in Section F, and therefore we lose the linear-time guarantee for solving the subproblem. To extend the linear-time core claim would further require low-rank approximations of kernel matrices $C$, e.g. via Nyström. We leave this as important future work outside the scope of this particular paper.

**Impact of imbalanced clusters.** Figure 12 shows the impact of imbalanced cluster sizes. In this example, the clusters contain approximately $10\%, 20\%, 20\%,$ and $50\%$ of the $n = 1000$ data points. We compared our method with NLR and spectral clustering (SC), omitting $K$-means++ due to its substantially higher error, which would distort the scale of the results. As the imbalance among clusters increases, we noted that the convergence becomes less stable with the default parameters, so we increase the rank $r$ from $K + 1$ to $K + 3$. The new imbalanced cluster size experiment further

reveals two things: (i) the performance of all methods under comparison degraded, and (ii) our method is the most robust (with the narrowest error distribution) among all.

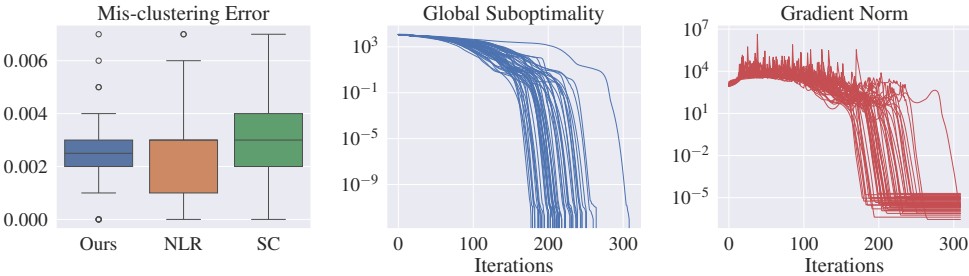

Figure 12: Clustering performance and convergence behavior when cluster sizes are imbalanced.

To explain this empirical observation, it is known that a similar (and more general) recovery threshold in the motivating $K$-means SDP to Equation (6) holds for unbalanced cluster sizes. In particular, the exact recovery threshold depends on the minimum of harmonic means of the cluster sizes, i.e. $m := \min_{1 \le k \neq l \le K}(2n_k n_l)/(n_k + n_l)$, in the following way (Chen & Yang, 2021b):

$$\overline{\Theta}^2 \gtrsim 1 + \sqrt{1 + \frac{d}{m \log n}}.$$

In the special case of $K$ equal clusters, $m = n/K$ attains the maximum possible value. Hence, when the cluster imbalance is significant, a larger centroid separation is required for the proposed algorithm to achieve clustering accuracy and convergence behavior comparable to that observed in the equal-cluster case.

**Runtime scaling with respect to $r$ and $d$.** Figure 13 shows the average per-iteration runtime as as $r$ and $d$ vary. The setting is the same as that of Figure 2, with $n = 1000$ fixed. The observed scalings for $r$ and $d$ are roughly $O(r^3)$ and $O(d)$, consistent with the leading terms of $O(nr^3(d+r) + r^6 + r^3 d^3)$. Because $n$ dominates as a leading constant, the contributions of the $r^6$ and $r^3 d^3$ terms are unlikely to become significant until $r$ and $d$ are extremely large.

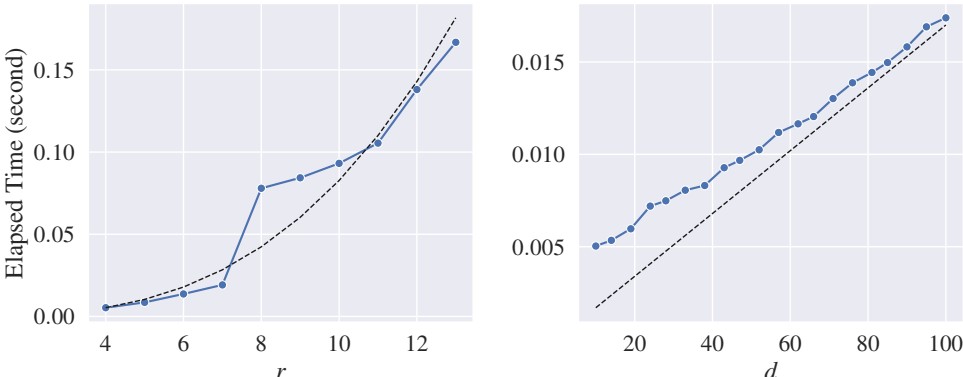

Figure 13: Average per-iteration runtime versus rank and dimension, $n = 1000$. The measured runtimes exhibit approximately $O(r^3)$ and $O(d)$ scaling (dashed lines).

**Robustness to initialization.** As illustrated by Figure 1, our method is robust to initialization, all 50 trials successfully converged to second-order optimal solutions. Although the solutions differ

(Figure 14), their corresponding membership matrices $Z$ are close to each other (Figure 15), and yield identical clustering result. Moreover, the minimum eigenvalues upon convergence form distinct clusters that align with clusters in the recovered membership matrix $Z$, as shown in Figure 15. These local critical points consistently produce perfect clustering, indicating that they remain close to the global optimum.

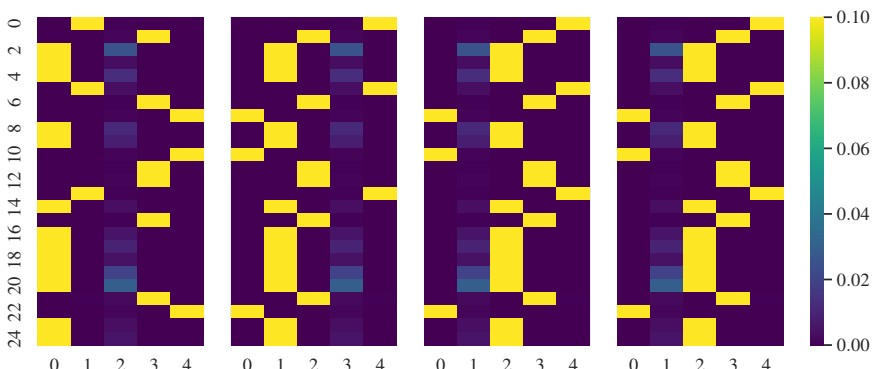

Figure 14: **Difference between the solutions.** First 25 rows of selected solution $U$ obtained from the global optimality experiment described in Section 4.

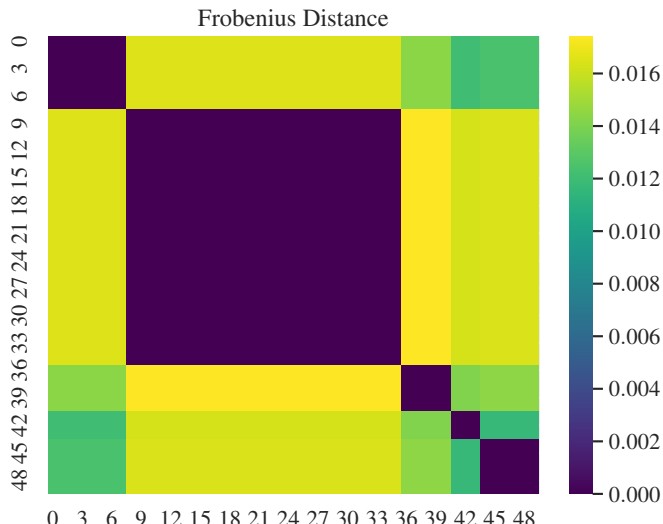

Figure 15: **Similarities of the membership matrices.** Frobenius distances between the membership matrices $Z$ obtained from the global optimality experiment in Section 4, sorted according to their corresponding minimum Hessian eigenvalues.

**Comparison with another Riemannian clustering method.** Inexact Accelerated Manifold Proximal Gradient Method (I-AManPG) by Huang et al. (2025) is a recent first-order Riemannian method for solving general problems of the form

$$\min_{X} \quad f(X) + \lambda\|X\|_1,$$
$$\text{s.t.} \quad X \in \mathcal{F}_v := \{X : X^\top X = I, v \in \text{span}(X)\}. \tag{41}$$

We evaluated its performance on the clustering problem using the on the CyTOF dataset with 50 repetitions (same setting as in Figure 3). The results are shown in Figure 16. While I-AManPG is

generally fast and accurate, its median error is higher than that of our methods. In particular, several runs of I-AManPG exhibited large errors, indicating convergence failures.

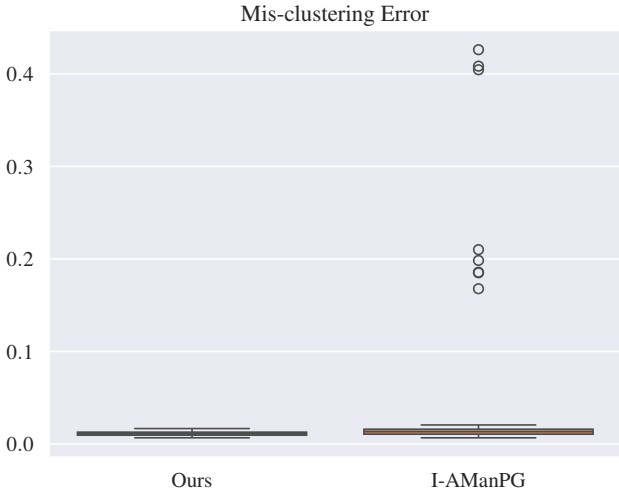

Figure 16: **Comparison with I-AManPG using CyTOF.** Performance of I-AManPG is comparable to other clustering methods. However, it suffers from convergence failures from time to time and requires careful tuning. Our method again demonstrated its accuracy and stability.

**Hyperparameters tuning.** In the various numerical experiments, we observed that a smaller value of $\mu$ led to more accurate solutions but at the cost of slower convergence. Therefore, we recommend selecting the largest possible $\mu$ that does not trigger the phase transition. A good heuristic we found is to choose such that the initial penalty term $\mu f_2$ remains less than 20 times the main term $f_1$ in the loss function. The onset of phase transition is also easy to notice, as the algorithm will quickly stagnate and terminate in just a few iterations. If higher accuracy is desired, one can reduce $\mu$ gradually, using the solution obtained with a larger $\mu$ as initialization. This warm-start strategy significantly speeds up convergence compared to using a small $\mu$ from the start.

As noted earlier, increasing $r$ can also improve accuracy, likely because it improves the problem landscape. However, due to the $\mathrm{poly}(r, d)$ runtime scaling, caution must be taken when deciding whether to decrease $\mu$ or increase $r$. If tuning $(\mu, r)$ together is desired, we suggest the following strategy: begin by decreasing $\mu$ with $r = K + 1$; if this does not produce satisfactory convergence or leads to excessive computation time, try increasing $r$ to $K + 2, \ldots, K + c$ for some small $c$, and repeat the $\mu$-tuning process.

The other hyperparameters in Algorithm 1 primarily influence the speed of the inner optimization. The initial multiplier $\lambda$ affects only the number of inner steps required during the first iteration. We recommend doing a simple trial run with only two iterations; the resulting optimization history typically offers a reliable guide for choosing an appropriate initial scale for $\lambda$. For the other two parameters, we suggest setting $\kappa_-$ slightly smaller than $\kappa_+$. Empirically, we found $\kappa_- = 1.1$ and $\kappa_+ = 1.3$ work well.

**Additional convergence plots.** Figure 17, Figure 18, and Figure 19 illustrate the convergence of our method on GMM with different parameters and on the CyTOF dataset, demonstrating its stability across different datasets.

**Hardware information.** All experiments in this work were conducted on a machine equipped with a single Intel Core i9-14900K CPU and 32 GB of RAM.

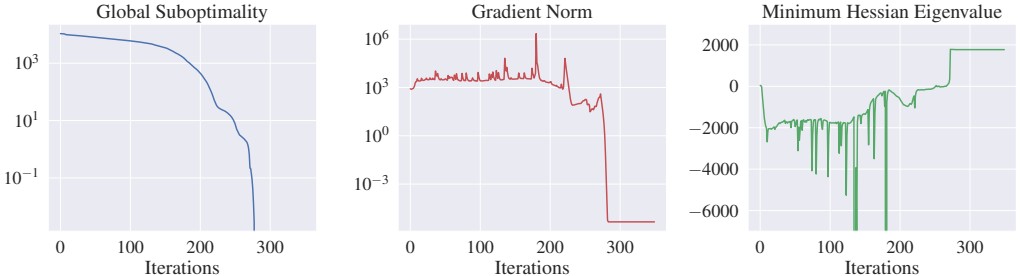

Figure 17: Convergence of our method on GMM data with perfect separation ($n = 500, \gamma = 1.0$). The loss value steadily decreases over iterations and converges rapidly near the optimal point. This example achieved a perfect final clustering result in the end.

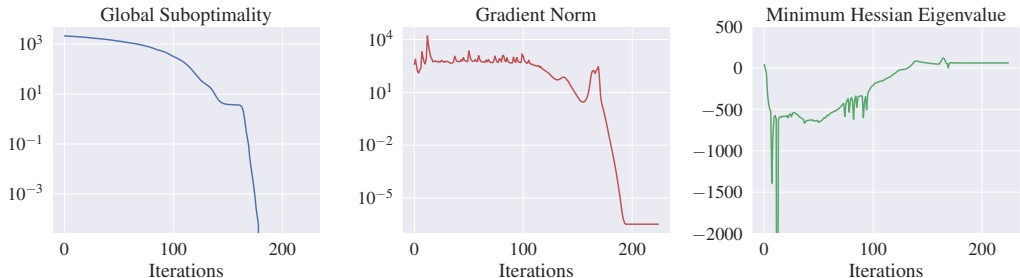

Figure 18: Convergence of our method on synthetic Gaussian mixture data with low separation ($n = 500, \gamma = 0.25$).

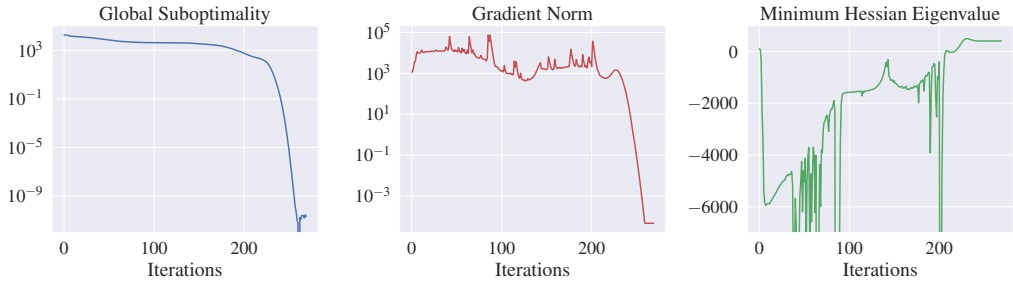

Figure 19: Convergence behavior of our method on the CyTOF dataset.

**License of assets.** The `MANOPT` solver is distributed under the terms of the GPLv3 license; the `PYMANOPT` solver is released under the 3-Clause BSD license; the CyTOF dataset is the work of Levine et al. (2015), cleaned and distributed by Weber (2015) under the MIT license.

**Code.** A Python demonstration of the proposed method is available at `https://github.com/Francis-Hsu/kmeans_manifold`. This repository provides example scripts for the GMM and CyTOF experiments presented in this work.

## H  PSEUDOCODE

This section presents the pseudocode of the proposed method. For its derivation, see Section F.

---

**Algorithm 1** Riemannian Second-order Method

---

**Require:** Data $X$,
  Initial point $(V_0, Q_0)$,
  Initial multiplier $\lambda$,
  Increment/decrement factors $(\kappa_+, \kappa_-)$,
  log-barrier penalty $\mu$,
  Max number of outer/inner iterations $T$ and $B$.
1: $(V, Q) \leftarrow (V_0, Q_0)$
2: **for** $i = 1, \ldots, T$ **do**
3:    Vectorize the input: $u \leftarrow [\text{vec}(V^\top), \text{vec}(Q)]^\top$;
4:    Compute the current loss: $\mathcal{L} = f(u)$;
5:    Compute the Riemannian gradient $G \leftarrow \nabla f(u)$;
6:    Compute the Jacobian $J \leftarrow \mathrm{D}\, g(u)$;
7:    Compute $\tilde{H} = \nabla^2_{uu} \mathcal{L}(u, y_u)$ as in Section F;
8:    **for** $j = 1, \ldots, B$ **do**
9:       $\dot{v}, \dot{q} \leftarrow \text{SOLVEINNER}(\tilde{H}, G, J, \lambda)$
10:       Reconstruct $\dot{V}, \dot{Q}$ from vector $\dot{v}, \dot{q}$
11:       $(V, Q) \leftarrow \mathrm{R}_{(V,Q)}(\dot{V}, \dot{Q})$
12:       Compute the new loss $\mathcal{L}'$ from $(V, Q)$
13:       **if** $\mathcal{L} > \mathcal{L}'$ **then**
14:          $\lambda \leftarrow \lambda / \kappa_-$
15:          **break**
16:       **else**
17:          $\lambda \leftarrow \lambda \cdot \kappa_+$
18: **output** $(V, Q)$
19:
20: **function** SOLVEINNER$(H, G, J, \lambda)$
21:    Add $\lambda I$ to the $V$ and $Q$ blocks of $H$;
22:    Form block matrices as in (40);
23:    $S \leftarrow L_{22} - L_{12}^\top D_{11}^{-1} L_{12}$
24:    $\begin{bmatrix} \dot{q} \\ z_0 \\ z_1 \end{bmatrix} \leftarrow \begin{bmatrix} b_2 \\ 0 \end{bmatrix} - L_{12}^\top D_{11}^{-1} b_1$
25:    $\dot{v} \leftarrow D_{11}^{-1} \left( b_1 - L_{12} \begin{bmatrix} \dot{q} \\ z_0 \\ z_1 \end{bmatrix} \right)$
26:    **return** $\dot{v}, \dot{q}$

---

