# OpenReview forum: "Scalable Second-order Riemannian Optimization for $K$-means Clustering"
_ICLR.cc/2026/Conference — ICLR 2026 Poster_

### Official Review · Reviewer_dYYy · 2025-10-28

**Soundness:** 3
**Presentation:** 4
**Contribution:** 3
**Rating:** 6
**Confidence:** 3

**Summary:**

The paper tackles the computational difficulty of solving the nonconvex, constrained formulation of K-means clustering. It reformulates the low-rank SDP relaxation as a smooth optimization over a Riemannian manifold, so that constraint feasibility is handled intrinsically. This allows the use of a cubic-regularized second-order Riemannian Newton method that guarantees convergence to second-order critical points—empirically aligned with global optima in K-means.

To make the approach scalable, the authors introduce a product-manifold factorization and exploit the Hessian’s structure to achieve linear-time per-iteration complexity. Experiments on Gaussian mixture and real-world datasets show faster convergence and lower mis-clustering rates than state-of-the-art first-order methods, demonstrating both theoretical soundness and strong empirical performance.

**Strengths:**

1. The paper successfully integrates second-order Riemannian methods, specifically the cubic-regularized Newton approach, into the K-means SDP framework. Moreover, The proposed product-manifold factorization resolves the computational bottleneck of expensive retraction operators in previous works.

2. Demonstrates consistent improvements in convergence speed and clustering accuracy over leading baselines, including NLR, RTR.

3. Sections 2 and 3 provides a concise review of SDP relaxations and information-theoretic thresholds. Theorem 1 and Theorem 2 are well-chosen to connect existing Riemannian convergence theory to this specific manifold formulation.

4. Reduces per-iteration complexity to linear in n, making second-order methods computationally competitive. The approach can be generalized to other manifold-constrained ML problems.

**Weaknesses:**

1. Assumption 1 (Benign Nonconvexity) remains unproven. The paper heavily relies on this assumption for its theoretical motivation but provides only empirical justification. A rigorous explanations or partial theoretical evidence would substantially strengthen the work.

2. In the numerical experiments, only a single real-world dataset (CyTOF) is used. Additionally, there is no ablation study on manifold dimension, sensitivity to initialization, or robustness to cluster imbalance.

3. The main theorems of complexity are adaptations of known Riemannian results rather than new convergence analyses specific to K-means.

4. The paper’s contribution seems somewhat incremental. It reformulates the K-means problem and employs a standard cubic-regularized Newton method, with the primary advancement lying in a more efficient computation of the Newton step by exploiting its structure.

**Questions:**

1. In line 295, how sensitive is the algorithm to the initial point $(V_0,Q_0)$?

2. Could this approach generalize to kernelized or probabilistic K-means variants? If so, how would the manifold and retraction structures adapt?

3. How is the second-order condition in Eq.(10) verified numerically? Does this influence empirical runtime?

4. As discussed in Appendix C.1, the Riemannian cubic-regularized Newton is equivalent to certain SQP methods. What are the computational advantages compared to projected Newton or augmented Lagrangian solvers?

5. In the proof of Lemma 4, the fourth equality is incorrect, but I believe it's just a typo and not a significant issue.

6. In Eq. (11), is the efficiency of the Hessian 1/2?

7. Typo on line 1127: this is not a quintic equation, but a quartic one.

---

> ### Author Response · Authors · 2025-11-25
> **Response to Reviewer dYYy (Weaknesses Part)**
>
> 1. Please see the global rebuttal for a full discussion of Assumption 1. In brief, while a formal proof would certainly strengthen the paper, current theory offers no path toward such a result in the nonnegative low-rank setting: every benign-landscape prototype in the literature becomes NP-hard once nonnegativity is imposed, so existing techniques cannot apply even in vastly simpler models. Partial results are possible only in a trivial neighborhood where the sparsity pattern is already correct, and so provides no insight into how the sparsity pattern is "discovered" by the algorithm. Within these limits, our extensive empirical evidence is consistent with how benign-landscape phenomena have historically first emerged in related fields, and it provides the strongest evidence currently attainable for this setting.
> 2. We thank the reviewer for these helpful suggestions. We have prepared an additional experiments on a new real-world dataset and on imbalanced data.
> To our understanding, the reviewer’s reference to “manifold dimension” concerns the rank parameter $r$. We already include an experiment in Section 4 (Figure 6) examining its impact, and we have updated our experiment to illustrate its effect when $\mu$ is mis-chosen. We also provide an experiment showing $r$'s impact on runtime.
> Robustness to initialization has been discussed in Appendix F and demonstrated in Figure 1.
> 3. The global rebuttal explains why developing convergence theory specific to $K$-means is inherently difficult. Any such analysis would require characterizing a *nontrivial neighborhood* large enough to capture how the algorithm transitions into the correct sparsity pattern. This is the same obstacle that prevents partial benign-landscape results: once the sparsity pattern is fixed, the problem becomes locally strictly convex and existing Riemannian results apply directly, but that region is trivial because it assumes the true clustering has already been identified. Establishing guarantees on the larger region needed to explain the algorithm’s behavior is essentially as hard as proving global benign nonconvexity, and no current techniques can handle this for nonnegative low-rank models. For this reason, our complexity guarantees follow the standard Riemannian framework, while the $K$-means-specific behavior necessarily remains empirical rather than theoretical.
> 4. We respectfully disagree with the characterization of the contribution as incremental. As detailed in the global rebuttal, the key advance is not the use of cubic regularization itself, but the structural insight that the shifted Hessian in our reformulation can be solved exactly in linear time, which fundamentally changes what is algorithmically possible. This insight enables every second-order method-*not just cubic regularization*-to run at the same per-iteration cost as a first-order method, something that was previously infeasible  for clustering or for broader nonnegative low-rank formulations.
> As validated in our experiments, first-order methods converge too slowly to be helpful due to the ill-conditioning brought on by the log barrier. In other words, if second-order methods could not be used (due to cubic-time cost), then there would be no practical way to solve our reformulation of the $K$-means problem. The whole idea of applying Riemannian optimization to $K$-means would fall apart.
> Of all possible second-order methods, we chose cubic regularization because it is one of the clearest and most demanding examples. Showing that even this method becomes cheap underscores the strength of the structure. Thus, the contribution is a new computational pathway for exact second-order optimization in clustering, rather than an incremental application of a known algorithm.

---

> ### Author Response · Authors · 2025-11-25
> **Response to Reviewer dYYy (Questions Part)**
>
> 1. Figure 1 shows an experiment verifying the robustness against initialization. See discussion in Appendix F.
> 2. Yes. The data enter the problem through the Gram matrix $C=XX^\top$ in (3), adapting the implementation for kernelized $K$-mean is straightforward and does not require modification on the manifold structures. We have updated our code to accept a kernel matrix as input, and included new experiments using the RBF kernel in the revised manuscript.
> Nevertheless, without direct access of the data matrix, we can no longer perform the sparse partition described in Appendix E, and therefore we lose the linear-time guarantee for solving the subproblem. To extend the linear-time core claim would further require low-rank approximations of kernel matrices $C$, e.g. via Nyström. We leave this as important future work outside the scope of this particular paper, where as assume a low-rank structure is provided *a priori* in the dataset and focus on how to leverage this to reduce computational cost.
> 3. For demonstration purposes, we explicitly computed all eigenvalues of the Riemannian Hessian, which is computationally expensive and has a noticeable impact on runtime. In practice, we perform this check only when the solution is nearly first-order optimal. Moreover, we use the inverse iteration $x \gets (H+\lambda I)^{-1}x$ to efficiently and reliably compute only the most negative eigenvalue, and then use a factorization test $H+\lambda I \succ 0$ to ensure that the $\lambda$ found is indeed the most negative.
> 4. Our reformulated $K$-means problem requires *strict feasibility* with respect to both nonnegativity and the simplex manifold, since the recovered factor must encode a valid partition. Our proposed formulation (3) maintains strict feasibility through the log-barrier and keeps iterates on the manifold through retraction, while the cubic-regularized Newton step guarantees rapid convergence to second-order critical points.
> In comparison, projected Newton methods are generally non-smooth, and lack robust guarantees, particularly when the feasible set is the intersection of a manifold with nonnegativity constraints. Moreover, projection onto this intersection can be difficult to compute reliably. Augmented Lagrangian methods, on the other hand, do not enforce strict feasibility at every iteration and may converge to points that are not second-order optimal for the original constrained problem.
> 5. 6. 7. Thank you for pointing out these errors. They have been fixed in the revised manuscript.

---

> ### Author Response · Authors · 2025-11-25
> **Additional Experiments**
>
> For convenience, we provide below the links to the plots presenting the results of the new experiments:
> - [Clustering with RBF kernel](https://imgur.com/kFNUVby)
> - [Clustering with imbalanced data](https://imgur.com/srjMWHm)
> - [Clustering performance when K is over-estimated](https://imgur.com/wwz1woK)
> - [Clustering performance when K is under-estimated](https://imgur.com/hk3TG0y)
> - [Real-world benchmark on the CIFAR10 data](https://imgur.com/IcCjFQr)
> - [Runtime benchmark w.r.t. rank and dimension](https://imgur.com/sbq9dCA)
> - [Updated Figure 6.](https://imgur.com/T9ksWi7)

---

### Official Review · Reviewer_dh1D · 2025-11-01

**Soundness:** 4
**Presentation:** 4
**Contribution:** 3
**Rating:** 8
**Confidence:** 3

**Summary:**

The paper proposes a new smooth, unconstrained Riemannian reformulation of a nonnegative low-rank factorization of the K-means SDP, enabling second-order optimization with feasibility preserved via retractions. The key idea is to map the original constraint set $\mathcal{M}$ onto a product manifold $\tilde{\mathcal{M}} = \mathcal{V}\times \mathrm{Orth}(r)$ via a submersion, which yields simple $O(nr + r^3)$ retractions and efficient expressions for Riemannian gradients/Hessians.

Building on this geometry, the authors design a cubic-regularized Riemannian Newton method that solves each Newton subproblem in $n\cdot\mathrm{poly}(r,d)$ time by exploiting a block-diagonal-plus-low-rank structure and a bisection scheme on the regularization parameter. The overall algorithm finds $\epsilon$–second-order points in $O(n\cdot\epsilon^{−3/2} \cdot\mathrm{poly}(r, d))$ time.

The work hinges on an empirical "benign nonconvexity" assumption: in regimes where the convex $K$-means SDP recovers ground truth, all approximate second-order critical points of the nonnegative low-rank model are near-global optima. The authors provide extensive empirical evidence for this behavior.

Experiments on synthetic GMMs and CyTOF data show that the proposed second-order method converges in hundreds of iterations, achieves similar or better clustering accuracy than the strongest first-order baseline (NLR), and substantially reduces time despite costlier iterations. It also outperforms prior Riemannian $K$-means methods and classical RTR/CG solvers that struggle with the log-barrier’s ill-conditioning.

The paper supplies detailed derivations: manifold geometry, LICQ verification, submersion proof, closed-form Lagrange multipliers for projections, efficient Hessian-vector products, feasible initialization (and necessity of $r > K$), and implementation details for the linear-time inner solves.

**Strengths:**

Conceptual advance: The submersion to a product manifold with simple retractions is elegant and practically impactful, removing the main bottleneck that hindered prior Riemannian approaches to $K$-means (expensive retractions and feasibility maintenance).

Algorithmic engineering: The cubic-regularized Riemannian Newton solver is carefully tailored—analytical gradients/Hessians, efficient tangent projections, and a block-diagonal-plus-low-rank exploitation for the inner linear systems. The bisection-based solver for the regularization parameter is simple and reliable.

Clear bridge to theory: The paper situates the contribution within the exact-recovery phase transition for the $K$-means SDP and explains how second-order points suffice under the benign nonconvexity hypothesis. The LICQ and manifold calculus are handled rigorously; the smoothing argument for the log penalty clarifies the use of second-order guarantees.

Strong empirical evidence: On both synthetic and real data, the method exhibits rapid, stable convergence to second-order points with near-zero mis-clustering; it consistently reduces iteration counts by orders of magnitude compared to NLR and RTR, translating to 2-4x faster runtimes despite more expensive steps.

Reproducibility and completeness: The paper provides explicit formulas, complexity accounting, initialization constructions, hyperparameter guidance, and ablations (rank $r$, penalty $\mu$, comparisons with several baselines), which make the contribution actionable for practitioners.

**Weaknesses:**

1. Assumption 1 (benign nonconvexity) is purely empirical in this work. While the authors motivate it with analogies to Burer–Monteiro and back it with experiments, the lack of any partial theoretical characterization (e.g., under separation/noise conditions and mild overparameterization $r > K$) limits the scope of the main claim in average-case regimes.

2. Dependence on the log-barrier: Although handled well algorithmically, the severe ill-conditioning induced by the log term drives both the design choices and some limitations (e.g., RTR/CG underperform). A discussion or experiment on alternative barrier/penalty designs (e.g., smooth-plus hard positivity via projections on $U$) could strengthen the case or show robustness.

3. Sensitivity to $\mu$ and feasibility interior: The method requires strictly positive $U$ and shows a phase transition when $\mu$ is too large. While the paper provides heuristics, a more systematic procedure (or adaptive schedule with safeguards) would make the solver more turnkey across datasets; also, the necessity of $r > K$ to ensure interior feasibility may be restrictive in memory-limited settings.

4. Scalability constants: The per-iteration complexity is linear in $n$ with $\mathrm{poly}(r, d)$ factors; however, the inner solves involve $r^3$ and $d$-dependent Schur complements. Scaling with varying $r$ and $d$ would help practitioners understand the limits and guide parameter choices.

5. Generality beyond GMM: Although the manifold formulation applies to kernelized $K$-means, empirical validation is focused on GMMs and one CyTOF setup. Broader tests (imbalanced/many clusters, higher $d$, other real datasets, kernels) would better support claims of robustness.

**Questions:**

1. Can you provide partial theory toward Assumption 1? For example, under the separation in Chen & Yang (2021b), and mild overparameterization $r = K + O(1)$, can you show absence of spurious second-order points in a neighborhood of the ground-truth factor, or establish a strict saddle property?

2. How robust is the method to mis-specified $K$ and to cluster imbalance? Could you include experiments where $K$ is under/over-estimated, and where cluster sizes vary significantly, and report both accuracy and convergence behavior?

3. Could you explore alternative penalties that retain positivity while improving conditioning (e.g., softplus smoothing, additive offsets, or barrier homotopy/continuation schedules), and compare convergence to your log-barrier?

4. What is the practical guidance for selecting $r$ beyond $r = K + 1$? Are there cases where larger $r$ improves optimization (escaping poor basins) or statistical robustness, and what is the runtime trade-off empirically?

5. Can you extend experiments to kernel $K$-means and non-Gaussian mixtures, or real vision/NLP datasets where the Gram matrix is built from learned features? This would help demonstrate the generality implied in Appendix A and the stated manifold framework.

---

> ### Author Response · Authors · 2025-11-25
> **Response to Reviewer dh1D**
>
> The following responses address both the weaknesses and the questions, organized according to the questions:
>
> 1. As noted in the global rebuttal, one can prove a partial result only in a very small neighborhood where the sparsity pattern is already correct; in that region, the objective is locally strictly convex, so there are no spurious second-order points and standard Newton theory gives superlinear convergence. But this neighborhood is trivial because it assumes the clustering has already been recovered. The real difficulty is to analyze a *nontrivial* region large enough to include the transition into the correct sparsity pattern under the Chen-Yang separation and mild overparameterization. Controlling this transition under inequality constraints and nonnegativity is essentially as hard as establishing global benign nonconvexity.
> Regarding strict saddles, this property is strictly stronger than benign nonconvexity (Assumption 1). Given the challenges outlined
> in the global rebuttal for even establishing Assumption 1, proving a strict saddle property is even further beyond current techniques.
>
> 2. We thank the reviewer for these suggestions. We have added more synthetic and real-world data experiments with mis-specified $K$ and imbalanced cluster size.
> To more rigorously explain the empirical observation, it is known that a similar (and more general) recovery threshold in the motivating $K$-means SDP to Eqn. (6) in the current paper holds for unbalanced cluster sizes. In particular, the exact recovery threshold depends on the minimum of harmonic means of the cluster sizes, i.e., $m := \min_{1 \leq k \neq l \leq K}(2 n_k n_l)/(n_k + n_l)$ in the following way [2]:
> $$
> \overline{\Theta}^2 \gtrsim  1 + \sqrt{1+ \frac{d}{m \log{n}}}.
> $$
> In the special case of $K$ equal clusters, $m = n / K$ and this configuration achieves the maximum of $m$. So, we expect that if the imbalance ratio is significant, larger centroid-separation is needed to achieve similar behavior of the proposed algorithm to clustering accuracy and convergence as the equal-cluster case.
>
> 3. We chose the log-barrier because it is a hard penalty that enforces strict feasibility along the manifold, and because it is self-concordant, which allows second-order methods to converge rapidly. While alternative barrier functions might improve convergence, they would not affect the core linear-time structural results. For this reason, exploring other barrier choices is beyond the scope of this paper.
> On the other hand, softplus and similar penalties do not guarantee strict positivity. Once an iterate becomes infeasible in the presence of manifold constraints, it is unclear how to "round" it back into the intersection of the manifold and the nonnegative orthant. Simple projection onto nonnegativity will typically move the point off the manifold and conflict with the retraction; enforcing both constraints simultaneously is highly nontrivial.
>
> 4. As indicated by Figure 6, when $r=K+1$ already yields a good solution, increasing $r$ has no obvious benefit. However, we have observed that when the algorithm becomes stuck, increasing $r$ can help it escape spurious regions. We modified Figure 6 to demonstrate this in the revised manuscript.
> Due to the $\operatorname{poly}(r, d)$ scaling, increasing $r$ has a significant impact on computation time. Therefore, if tuning $(\mu,r)$ together is desired, we suggest the following strategy: begin by decreasing $\mu$ with $r=K+1$; if this does not produce satisfactory convergence or leads to excessive computation time, try increasing $r$ to $K+2,\dotsc,K+c$ for some small $c$, and repeat the $\mu$-tuning process.
> We have also added a new experiment showing the runtime trade-off.
>
> 5. Since the data enters the problem through the Gram matrix $C=XX^\top$ in (3), adapting the implementation is conceptually straightforward for extension to the kernel K-means setting. We have modified our code to accept a kernel/Gram matrix as input, and added experiments using the RBF kernel in the revised manuscript.
> However, here we want to point out a computational overhead for the kernel $K$-means. Without direct access of the data matrix, we can no longer perform the sparse partition described in Appendix E, and therefore we lose the linear-time guarantee for solving the subproblem. To extend the linear-time core claim would further require low-rank approximations of kernel matrices $C$, e.g. via Nyström. We leave this as important future work outside the scope of this particular paper, where as assume a low-rank structure is provided a priori in the dataset and focus on how to leverage this to reduce computational cost.
> Finally, we also added a second real-data example using high-dimensional embeddings of CIFAR-10. Our observation is largely consistent with the CyTOF data.
>
> References:
> - [1] Chen and Yang. (2021) Cutoff for Exact Recovery of Gaussian Mixture Models. IEEE Transactions on Information Theory. 67(6), 1557-9654.

---

> ### Author Response · Authors · 2025-11-25
> **Additional Experiments**
>
> For convenience, we provide below the links to the plots presenting the results of the new experiments:
> - [Clustering with RBF kernel](https://imgur.com/kFNUVby)
> - [Clustering with imbalanced data](https://imgur.com/srjMWHm)
> - [Clustering performance when K is over-estimated](https://imgur.com/wwz1woK)
> - [Clustering performance when K is under-estimated](https://imgur.com/hk3TG0y)
> - [Real-world benchmark on the CIFAR10 data](https://imgur.com/IcCjFQr)
> - [Runtime benchmark w.r.t. rank and dimension](https://imgur.com/sbq9dCA)
> - [Updated Figure 6.](https://imgur.com/T9ksWi7)

---

### Official Review · Reviewer_9AoF · 2025-11-03

**Soundness:** 3
**Presentation:** 4
**Contribution:** 3
**Rating:** 6
**Confidence:** 5

**Summary:**

This paper tackles the nonconvex low-rank factorization of SDP relaxation of K-means clustering. By penalizing the nonnegative constraints, the authors view the resulting problem as a manifold optimization problem and apply the Riemannian cubic regularized Newton to obtain a 2nd-order solution.

**Strengths:**

1. Develop a fast algorithm from manifold optimization viewpoint (Theorem 2), which is novel at least to me.
2. The subproblems of Riemannian cubic regularized Newton can be solved efficiently.
3. Empirically, 2nd-order critical points are globally optimal.

**Weaknesses:**

1. This paper only provides sublinear convergence rate. Is it possible to prove stronger local convergence rate results? For example, quadratic/superlinear convergence rate. Fig1 shows fast convergence rate in practice.
2. The benign nonconvexity is described in Assumption 1. Is it possible to say something about benign landscape? For example, related work of benign nonconvexity results.

**Questions:**

1. Is the part of lines 311-355 new result? If this is not new, it would be better to cite some references.
2. Could the authors explicitly explain why Riemannian cubic regularized Newton can overcome the ill-conditioning and RGD/RTR cannot?

---

> ### Author Response · Authors · 2025-11-25
> **Response to Reviewer 9AoF**
>
> ### Weaknesses
> 1. A stronger local convergence result is possible, but only in the trivial neighborhood where the correct sparsity pattern has already been identified. In that region, the problem becomes locally strictly convex, so any standard second-order method enjoys superlinear or even quadratic convergence. As discussed in the global rebuttal, the real challenge is to prove a nontrivial neighborhood large enough to explain the rapid convergence *to the right sparsity pattern* as observed in Fig. 1. Unfortunately, establishing such a result is much harder; it is essentially as hard as proving a benign landscape globally, so is far beyond current techniques.
>
> 2. Please see the global rebuttal on Assumption 1 for a full explanation of why a rigorous benign-landscape guarantee is currently beyond reach in any nonnegative low-rank setting. In brief: all known benign-landscape results rely on prototypes that collapse to NP-hard problems under nonnegativity, so existing proof techniques cannot apply here. Within these limits, our contribution is the strongest possible: if the landscape is benign (as our extensive experiments strongly indicate) then any second-order point yields the optimal clustering. This conditional reasoning is standard in early benign-landscape work (for example, matrix completion or phase retrieval before 2016, and many modern deep-learning models today).
>
> ### Questions
> 1. This section is essentially a Riemannian extension of known results on the approximate minimization of cubic-regularized subproblems, and aligns closely with Section 6.1 of [3]. The underlying ideas are nearly identical; the analysis simply takes place in the tangent space rather than in an Euclidean space. We will make this explicit in the revised version.
>
> 2. Ill-conditioning affects first-order methods that rely only on gradient information. RGD is purely first-order, and RTR, although nominally second-order, solves its inner subproblems with truncated conjugate-gradient, which is also a first-order method. In contrast, Riemannian cubic-regularized Newton is a truly second-order method; at each step, the shifted Hessian $(H+\lambda I)^{-1}$ acts as the *optimal* preconditioner, fully compensating the effects of ill-conditioning. In generic problems this is too expensive, but in our reformulation the shifted Hessian can be solved exactly in linear time, so the algorithm is able to take a true second-order step at essentially first-order cost.
>
> ### References
> - [1] Rebjock \& Boumal. (2024). Optimization over bounded-rank matrices through a
> desingularization enables joint global and local guarantees.
>
> - [2] Carmon et al. (2011). Lower Bounds for Finding Stationary Points I.
>
> - [3] Cartis et al. (2011). Adaptive cubic regularisation methods for unconstrained optimization. Part I: motivation, convergence and numerical results.

---

### Official Review · Reviewer_9Enk · 2025-11-04

**Soundness:** 3
**Presentation:** 3
**Contribution:** 2
**Rating:** 4
**Confidence:** 3

**Summary:**

This paper proposes a scalable second-order Riemannian optimization algorithm for solving the K-means clustering problem through a low-rank semidefinite relaxation. The authors reformulate the nonconvex problem as a smooth unconstrained optimization problem over a product manifold and apply a cubic-regularized Riemannian Newton method. The algorithm is claimed to achieve linear per-iteration complexity with provable convergence to second-order critical points.

However, several issues remain. The key assumption (Assumption 1) seems inconsistent and lacks justification; the algorithmic novelty is limited.   If the authors can address my concern, I would raise my score.

**Strengths:**

- The mapping from the constraint manifold $\mathcal{M}$ to the product manifold provides a principled and novel way to represent K-means as a smooth manifold problem.   This reformulation reduces the projection cost from $O(n^2)$ to $O(nr + r^3)$.
- By exploiting block-diagonal-plus-low-rank Hessian structure (Appendix E), each iteration scales linearly in \(n\) in the cubic-regularized Newton approach.
- Use bisection search to solve the subproblem in the cubic-regularized Newton approach.

**Weaknesses:**

- Assumption 1 is difficult to interpret. The problem (1)  does not require \(U > 0\),  (U > 0\) enforced in problem (3), which is only a sufficient but not necessary condition for $Z = UU^\top > 0$. Hence, the assumption is logically inconsistent with the model definition.  There exists the case that the optimal solution $U_{ij}<0$ .  In addition, this assumption is verified only empirically (Fig. 1), with no analytical justification or example provided.  Since all global optimality claims rely on this assumption, its ambiguity weakens the theoretical contribution.

- The method essentially applies a standard cubic-regularized Newton algorithm to a reformulated manifold problem, which not  introduce new algorithm, or adaptive strategies, or theoretical improvements.

- Although the reformulation simplifies the constraint structure, computation of gradients and Hessians remains costly.  I recommend the authors discuss hybrid methods that separate simple manifold constraints (handled via Riemannian optimization) and other remained constraints (handled via augmented Lagrangian function).  Related work worth discussing includes:

  - Wang, J., & Hu, L. (2025). Solving low-rank semidefinite programs via manifold optimization. Journal of Scientific Computing, 104(1), 33.

  - Monteiro, R. D., Sujanani, A., & Cifuentes, D. (2024). A low-rank augmented Lagrangian method for large-scale semidefinite programming based on a hybrid convex-nonconvex approach. arXiv preprint arXiv:2401.12490.
  -  Hou, D., Tang, T., & Toh, K. C. (2025). A low-rank augmented Lagrangian method for doubly nonnegative relaxations of mixed-binary quadratic programs. Operations Research.
  -  Wang, Y., Deng, K., Liu, H., & Wen, Z. (2023). A decomposition augmented lagrangian method for low-rank semidefinite programming. SIAM Journal on Optimization, 33(3), 1361-1390.

- Important components—algorithm pseudocode, subproblem solvers, and retraction details—are placed only in the appendix.

**Questions:**

Why is the cubic-regularized Newton framework preferred? The original method (Agarwal et al., 2021) solves subproblems via *Lanczos iterations*, whereas this paper adopts bisection search. The authors should discuss those two approaches.

---

> ### Author Response · Authors · 2025-11-25
> **Response to Reviewer 9Enk (Weaknesses Part)**
>
> 1. Thanks for your comments. We clarify the relationship between the original $K$-means problem (1), its standard SDP relaxation (2), and our proposed nonnegative matrix factorization (NMF) formulation (3), and how Assumption 1 should be interpreted.
>     1. **Problem (1) does require $U\\ge0$.** As shown in [CMVW17],
> the combinatorial $K$-means problem (1) is exactly equivalent to:
> $$
> \\min\_{U\\in\\mathbb{R}^{n\\times K}}\\{\\langle-XX^{\top},UU^{\top}\\rangle:U\geq0,UU^{\top}\mathbf{1}_{n}=\mathbf{1}\_{n},U^{\top}U=I\_{K}\\} .
> $$
> Thus, nonnegativity is inherent to $K$-means; it is not an additional
> modeling assumption introduced by (3). However, the above problem does not admit a strictly feasible interior and so is not amenable to modern optimization tools.
>     2. **Our NMF formulation (3) is a tight relaxation of this equivalent problem in the exact-recovery regime.** We replace the hard constraint $U\ge0$ with a log barrier, and then relax $U^{\top}U=I_{K}$ into $\operatorname{tr}(UU^{\top})=K$ to yield
> $$
> \min\_{U\in\mathbb{R}^{n\times r}}\Bigl\\{\langle-XX^{\top},UU^{\top}\rangle+\mu\sum_{i,j}\log(U_{i,j})\_{+}:UU^{\top}1=1,\,\mathrm{tr}(UU^{\top})=K\Bigr\\}.
> $$
> This problem does indeed have a strictly feasible interior. As we now explain, the relaxation from (1) to (3) is tight under
> standard exact-recovery separation conditions, meaning that its global
> minima match the true clustering.
>     3. **The classic SDP relaxation (2) is weaker, but becomes tight in the exact-recovery regime.** As the reviewer notes, the containment $\\{UU^{T}:U\ge0\\}\\subset\\{Z:Z\succeq0,Z\ge0\\}$
> is strict in general, so replacing $UU^{T}$ by $Z$ yields a relaxation.
> [CY21] show that the resulting relaxation from (1) to (2) is exactly
> tight above the statistical separation threshold: its optimizer factors
> as $Z=UU^{\top}$ with $U\ge0$, and the sparsity pattern of $U$
> recovers the true cluster labels. Since (3) is tighter than (2), it
> must also be tight in this regime.
>     4. **Assumption 1 concerns only the nonconvex landscape of (3), not its modeling validity.** Once tightness holds, the global optima of (3) match those of (1) and (2). Assumption 1 simply conjectures (supported by strong experiments) that the only second-order critical points of (3) lie near these optima. This explains the empirical reliability of our second-order method.
>     - [CMVW17] Carson, Mixon, Villar, and Ward. "Manifold optimization for k-means clustering.'' IEEE SampTA 2017.
>     - [CY21] Chen and Yang. "Cutoff for exact recovery of gaussian mixture models.'' IEEE Trans. Info. Theory 2021.
>
> 2. Please see the global rebuttal on Assumption 1 for a full explanation of why a rigorous benign-landscape guarantee is currently beyond reach in any nonnegative low-rank setting. In brief: all known benign-landscape results rely on prototypes that collapse to NP-hard problems under nonnegativity, so existing proof techniques cannot apply here. Within these limits, our contribution is the strongest possible: if the landscape is benign (as our extensive experiments strongly indicate) then any second-order point yields the optimal clustering. This conditional reasoning is standard in early benign-landscape work (for example, matrix completion or phase retrieval before 2016, and many modern deep-learning models today).
> 3. As explained in the global rebuttal, our contribution is structural, not algorithmic. Once the reformulation is in place, the shifted-Hessian system becomes exactly solvable in linear time, so *any* second-order method can be run at the same per-iteration cost as a first-order method. Cubic regularization is simply the clearest demonstration: it has one of the heaviest subproblems, so showing that even this method becomes cheap highlights the strength of the structure. The advance is enabling exact second-order methods for clustering, which was previously infeasible.
> 4. Thank you for these pointers, which we will cite in the revision. The cited methods target low-rank SDPs where feasibility can be relaxed and corrected by augmented-Lagrangian steps. But our setting is fundamentally different. Our reformulated $K$-means problem requires *strict feasibility* with respect to both nonnegativity and the simplex manifold, since the recovered factor must encode a valid partition. Augmented-Lagrangian or projection-based schemes do not preserve this property and break the structure we rely on.
> On the other hand, we disagree with the reviewer's assertion that "computation of gradients and Hessians remains costly''. Our main technical result is precisely that Riemannian gradients and Hessians can be computed in linear $O(n)$ time. The same structural ideas could potentially reduce the costs inside the cited SDP methods to linear $O(n)$ time. Of course, different algorithms may have different leading constants and convergence properties, so determining the best practical choice is a natural direction for future work.

---

> ### Author Response · Authors · 2025-11-25
> **Response to Reviewer 9Enk (Questions Part)**
>
> As discussed in the global rebuttal, cubic regularization is chosen as a simple illustration and not necessarily the ``best'' algorithm. Our structural insight is the main contribution, not the solver choice. The same insight also enables exact linear-time implementations of other second-order methods.
>
> Regarding Lanczos: in the general setting of Agarwal et al. (2021), solving the shifted Hessian equation $(H+\lambda I)x=b$ exactly cost cubic $O(n^{3})$ time so is too expensive. Instead, they use Lanczos to build a low-dimensional Krylov subspace, projects the shifted Hessian equation into that subspace, and then performs bisection search on the projected cubic equation (as in [CGT21, Sec 6.1]). This yields only an *approximate* step, exact only within the Krylov subspace.
>
> In our setting, the shifted-Hessian system can be solved exactly in linear $O(n)$ time. This removes the need for a Krylov approximation, so we directly apply the full bisection procedure of [CGT21, Sec 6.1] and obtain the true cubic-regularization step without projection. The principle is the same, but our structure eliminates the approximation stage and produces an exact step.
> - [CGT21] Cartis, Gould, Toint. "Adaptive cubic regularisation
> methods for unconstrained optimization. Part I: motivation, convergence
> and numerical results." Math. Program. 2011.

---

### Author Response · Authors · 2025-11-25
**Global Response, Part 1**

We thank the reviewers for their thoughtful comments. The main concerns across reviews focus on two questions: the role of Assumption 1 in light of the lack of a formal benign-landscape guarantee, and whether our use of cubic regularization should be viewed as an incremental
application of known techniques. Before responding to these issues in detail, we restate the contributions of the paper to give a clear perspective on what the work achieves.

### Clarifying the contributions of the paper
This paper makes three key contributions:

**1. A continuous reformulation of combinatorial $K$-means.** We show that under standard exact-recovery separation conditions, the
combinatorial $K$-means problem admits an equivalent smooth nonconvex formulation with a strictly feasible interior whose global minima match the optimal clustering. This makes the problem accessible to modern optimization tools and grounds the rest of the paper.

**2. Empirical evidence for an unexpectedly well-behaved landscape.**
In the exact-recovery regime, we observe strong empirical evidence that greedy local optimization converge reliably to the global optimum, despite the nonconvex nature of the problem. We present extensive experiments supporting this behavior, which motivates Assumption 1. Combined with Contribution 1, the assumption reduces $K$-means into the much easier problem of computing a second-order stationary point of a smooth nonconvex objective.

**3. Exact second-order order methods can be made as cheap as first-order methods.** Typical nonconvex problems force a tradeoff between cheap but slow first-order methods (like gradient descent and CG-RTR) and fast but expensive exact second-order methods (like cubic reglarization). Unusually for our reformulation, problem structure allows exact second-order steps, including \emph{but not limited to} cubic regularization, to be carried out with the same per-iteration cost as first-order methods. In this setting, exact second-order methods dominate, since they converge much faster while asymptotically costing the same per iteration.

### Global benign landscape (Assumption 1)
Several reviewers ask whether the well-behaved landscape we observe empirically can also be justified in theory.  While we provide extensive experiments, a full guarantee would require tools that do not yet exist. The literature contains three main families of benign landscape results, but each relies on structural features that fail under nonnegativity $U\ge0$:
- **Unconstrained minimization under well-conditioning assumptions [BMS16, GJZ17, Zha24].** The core insight is that the idealized function $f_{0}(U)=\\|UU^{T}-M\\|\_{F}^{2}$ is benignly nonconvex over $U\\in\\mathbb{R}^{n\times r}$, so well-conditioned perturbations inherit this behavior. Once we impose $U\ge0$, minimizing $f_{0}$ becomes equivalent to nonnegative matrix factorization, which is NP-hard, so the prototype behind these results is no longer benign in the nonnegative setting.
- **Minimization subject to smooth manifold constraints [BVB20].** In the unconstrained low-rank case, every rank-deficient second-order point is globally optimal, and with enough overparameterization all second-order points become rank-deficient. Under nonnegativity $U\ge0$, rank deficiency no longer guarantees optimality, so this mechanism breaks.
- **Group synchronization and related quadratic models [Lin23, MB24].** These results rely on the benign landscape in the spectral relaxation $\min\langle C,UU^{T}\rangle$ subject to $\mathrm{tr}(UU^{T})=1$. Under nonnegativity, this prototype becomes equivalent to copositive testing, which is NP-hard. Thus, the base case for this line of work also fails.

Taken together, benign landscape theory currently covers a narrow collection of structured problems, each reducible to a well-behaved unconstrained prototype. In every case, the nonnegative counterpart is non-benign, so existing theory cannot extend to our setting. A proof of Assumption 1 would require (i) identifying a nonnegative low-rank prototype that actually has a benign landscape, and (ii) reducing our formulation to it. To our best knowledge, no such prototype exists for nontrivial $r>1$.

To summarize, a rigorous guarantee for Assumption 1 is well beyond current theory. In related areas like matrix sensing, matrix completion, and phase retrieval, empirical evidence of benign behavior long preceded any formal analysis. Our experiments follow this pattern and provide the strongest evidence currently available for this constrained setting.

---

### Author Response · Authors · 2025-11-25
**Global Response, Part 2**

### Local benign landscape?
Reviewers further ask whether a more modest local guarantee might be feasible. One can indeed show a tiny local result: once the iterates already match the correct sparsity pattern (that is, once the locations of near-zero entries $U_{i,j}\approx0$ stabilize), the objective becomes locally strictly convex and standard second-order methods converge superlinearly. But this neighborhood is trivial, since it assumes the clusters have already been recovered.

The real challenge is a nontrivial neighborhood large enough to include the transition into the correct sparsity pattern. Proving such a result
is essentially as hard as a global guarantee, because it requires explaining how the algorithm discovers the correct support. Existing
results of this form appear only in much simpler unconstrained models and required major advances there [MWCC18, SS21]. Our Riemannian, inequality-constrained, nonnegative formulation is much more complex. Understanding how sparsity patterns emerge in log-barrier problems remains completely open.

### On our use of Cubic Regularization
Reviewers 8Enk and dYYy suggest that our use of cubic regularization makes the work incremental, since it applies a standard second-order method to a reformulated problem. We believe this impression comes from focusing on the choice of algorithm rather than the structural observation that makes such algorithms practical.

Our key insight is that, for our reformulation, the shifted-Hessian system $(H+\lambda I)x=b$ can be solved in $O(n)$ time, where $H$ is the Riemannian Hessian. With this structure, every second-order method---not just cubic regularization---admits an exact linear-time implementation. This includes all of the second-order methods cited by the reviewers; each can be upgraded to an exact subproblem solve
with the same per-iteration cost.

We chose cubic regularization not because we endorse it as the best method, but because it is simple to explain and has one of the most
demanding subproblems. Showing that even this algorithm can be implemented cheaply illustrates the strength of the structural insight. More sophisticated second-order methods may further reduce per-iteration cost by using easier subproblems or relaxed exactness. Exploring such variants is a natural direction for future work.

References:
- [BNS16] Bhojanapalli, Neyshabur, and Srebro. "Global optimality of local search for low rank matrix recovery.'' NIPS 2016.
- [GJZ17] Ge, Jin, and Zheng. "No spurious local minima in nonconvex low rank problems: A unified geometric analysis.'' ICML 2017.
- [Zha24] Zhang. "Improved global guarantees for the nonconvex burer--monteiro factorization via rank overparameterization.'' Math. Program. 2024.
- [BVB20] Boumal, Voroninski, and Bandeira. "Deterministic Guarantees for Burer‐Monteiro Factorizations of Smooth Semidefinite Programs.'' Comm. Pure Appl. Math. 2020.
- [Lin23] Ling. "Solving orthogonal group synchronization via convex and low-rank optimization: Tightness and landscape analysis.'' Math. Program. 2023.
- [MB24] McRae and Boumal. "Benign landscapes of low-dimensional relaxations for orthogonal synchronization on general graphs.'' SIAM J Optim. 2024.
- [MWCC18] Ma, Wang, Chi, Chen. ``Implicit regularization in nonconvex statistical estimation: Gradient descent converges linearly for phase retrieval and matrix completion.'' ICML 2018.
- [SS21] Stöger and Soltanolkotabi. ``Small random initialization is akin to spectral learning: Optimization and generalization guarantees
for overparameterized low-rank matrix reconstruction.'' NeurIPS 2021.

---

### Meta-Review · Area_Chair_LJvY · 2025-12-26

**Summary:**

This paper introduces a new Riemannian relaxation of the K-means clustering problem. A main technical contribution is that the Riemannian manifold can be written as the submersion of the product of two simpler manifolds. This observation greatly simplifies the subproblem involved in a Riemannian cubic regularized Newton method for solving the Riemannian optimization. Numerical experiments are conducted to demonstrate the effectiveness of the proposed approach. Overall, this paper provides a nice contribution to solving K-means clustering. One downside of the paper is that the Assumption 1 is not justified theoretically. But given that other contributions in the paper are significant, addressing this issue can be left to a future work.

**Reviewer Concerns:**

Addressed: why choosing Riemannian cubic regularized Newton method.

Outstanding: theoretically justify Assumption 1.

**Reviewer Scores:**

I think the authors satisfactorily explained why they chose the Riemannian cubic regularized Newton method. The authors also provided some insights why theoretically justify Assumption 1 can be difficult. Reviewers such as 9Enk, 9AoF, dYYy might increase their scores if discussions can be enabled.

---

### Decision · Program_Chairs · 2026-01-26

Accept (Poster)